# SpeechQC-Agent: A Natural Language Driven Multi-Agent System for Speech Dataset Quality

## Abstract

Ensuring the quality of large-scale datasets is a prerequisite for reliable machine learning, yet current verification pipelines are static, domain-specific, and heavily reliant on human experts. We introduce **SpeechQC-Agent**, the first natural language-driven *agentic framework* for dataset quality control that generalizes across modalities, vendors, and languages. A central planner LLM decomposes user queries into directed acyclic graph (DAG) workflows executed by modular sub-agents that combine reusable tools with LLM-synthesized functions, enabling flexible and scalable verification. Unlike rule-based scripts, this design supports parallelism, dependency management, and adaptive extension to novel schemas. To benchmark verification systems, we release **SpeechQC-Dataset**, a multilingual speech corpus with controlled perturbations spanning audio, transcripts, and metadata, allowing systematic evaluation of 24 verification tasks. Experiments show that SpeechQC-Agent achieves 80-90% of expert-level accuracy while operating at less than 20% of the cost and time, and generalizes from synthetic perturbations to real vendor-supplied corpora. Comparative analysis across multiple planner LLMs highlights trade-offs between fidelity (GPT-4.1-mini), efficiency (LLaMA-3.3-70B), and reasoning strength (DeepSeek-R1). Beyond speech, our approach establishes a general paradigm for LLM-driven workflow generation in dataset quality assurance, with implications for the curation of multimodal and multilingual resources on scale.

## 1 Introduction

India is the epicenter of linguistic diversity (Graziosi, 2017), and the Census of India(India, 2001) reported 30 languages spoken by more than a million native speakers. However, despite this diversity, even widely spoken languages such as Hindi (Javed et al., 2024a) remain under-resourced in the context of publicly available speech-text datasets. Building speech technologies such as Automatic Speech Recognition (ASR) (Kumar et al., 2022b), Text-to-Speech (TTS) (Tankala et al., 2024), Speech Translation (ST) (Gupta et al., 2025), etc., for these languages is critically dependent on the availability of large-scale, high-quality, and diverse speech datasets (Javed et al., 2023). However, curating such datasets is a slow, labor-intensive process fraught with several challenges. For example, manual verification for 1,000 hours of conversational speech may require 3-4 annotators working full-time for 6 months, making scalability prohibitive (Kumar et al., 2022a).

Speech dataset construction typically involves collaboration with multiple vendors, each following different conventions for audio encoding (e.g., sampling rate, file format, channel configuration), transcript formatting (e.g., CSV vs. JSON, sentence vs. file-level alignment), and metadata organization (e.g., speaker demographics or dialect tags) (Javed et al., 2024b). This heterogeneity makes it difficult to design unified processing pipelines. Beyond formatting inconsistencies, dataset quality requires extensive manual validation: transcripts must be checked for accuracy, audio must be screened for corruption or poor recording conditions, and speaker demographics must be monitored to maintain linguistic and social diversity. In practice, such validation is either performed through random sampling (He et al., 2024), which is fast but risks overlooking systemic errors, or by exhaustive verification (Jiang et al., 2024), which ensures quality but is prohibitively slow. These

Figure 1: Our system leverages structured prior knowledge and parallel planning capabilities to generate efficient, self-managing task workflows for speech dataset verification.

challenges make high-quality corpus creation both time-intensive and resource-demanding (Kumar et al., 2025), underscoring the need for scalable, automated verification frameworks.

Several initiatives have sought to address the scarcity of high-quality, standardized, and scalable speech datasets for Indian languages. Projects such as AI4Bharat (Javed et al., 2024b), IITB (Adiga et al., 2021) and the Vaani (Team, 2025) collaboration represent important steps toward resource creation, but they remain constrained by human-in-the-loop verification pipelines. Recruiting and training large annotator teams is logistically complex and financially burdensome, and manual checks do not scale reliably as dataset size or vendor diversity increases. Similarly, the Spring Lab (Sarkar et al., 2025) random-sampling approach has been shown to overlook recurring systematic error types in speech datasets, reducing their reliability for downstream applications. As a result, no existing solution provides a scalable and automated framework that simultaneously ensures efficiency and high quality in the verification of multilingual speech dataset.

In parallel, recent advances in Large Language Models (LLMs) as agents (Guo et al., 2024) have demonstrated competitive performance in tool use (Li et al., 2023), planning (), and decision-making (Yao et al., 2023) tasks. Although these advances have transformed many areas, their potential in speech dataset quality control, a relatively niche domain, remains largely untapped. The scarcity of specialized models and benchmarks in this area is due to two key limitations: (i) the absence of comprehensive, high-quality datasets that capture diverse real-world error conditions, and (ii) the heterogeneity of speech-text data formats across languages and vendors. Moreover, prior agentic systems designed for text-based data processing face well-documented challenges: inconsistent environment configurations (Hu et al., 2024b), difficulty adapting to novel schemas (Tang et al., 2023), hallucinating actions (Zhong et al., 2024b), unnecessary repetition of steps (Zhang et al., 2024a), and weak contextual grounding (Song et al., 2023). These problems are magnified in multimodal speech settings, where alignment between audio, transcripts, and metadata is critical and difficult to verify.

In this paper, we introduce SpeechQC-Agent, a natural language-driven multi-agent framework for automating the quality control and verification of large-scale speech datasets. Unlike prior approaches that rely on fixed scripts or manual annotator checks, SpeechQC-Agent leverages a centralized LLM to interpret user instructions and orchestrate specialized sub-agents for format normalization, transcript validation, audio quality checks, and metadata verification. By allowing users to issue natural language prompts (for example, "Check the audio files for sample rate, corruption, and domain"), the system dynamically constructs task-specific workflows, reducing human dependency and enabling scalable dataset processing (Figure 1). Beyond synthetic perturbations, we validate SpeechQC-Agent on vendor-supplied corpora, showing that the framework generalizes to real-world noise and annotation inconsistencies.

This paper makes the following key contributions:

1. **Natural Language-Driven Workflow Generation:** We present the first system to automatically generate speech dataset verification workflows directly from natural language prompts, reducing the dependency on rigid rules or manual scripting.

2. **Modular Multi-Agent Execution Framework:** We propose a graph-based framework that decomposes verification into modular sub-agents, enabling both task-level parallelism and structured

dependency management.

3. **Tool Synthesis and Reuse:** We demonstrate how LLM-synthesized tools can be combined with robust pre-defined components (e.g., VAD, domain identification, CTC scoring), supporting both adaptability and efficiency.

4. **First Application to SpeechQC-Dataset:** We release **SpeechQC-Dataset**, a synthetic yet realistic multilingual benchmark, and show that **SpeechQC-Agent** is the first end-to-end system capable of applying agentic workflow generation to real-world speech-text data quality control across heterogeneous vendor formats[1].

## 2 RELATED WORK

Recent advances in large language models (LLMs) have enabled agent-based frameworks for automated task orchestration and tool-driven reasoning (Liu et al., 2023; 2024; Zhong et al., 2024a; Zhu et al., 2024; Sun et al., 2024; Xie et al., 2024). Our work intersects with three lines of research: (i) LLM-powered multi-agent collaboration, (ii) workflow generation and evaluation, and (iii) modular agent design. However, none directly address the verification of large-scale **speech datasets**, particularly in low-resource multilingual settings.

MacNet (Qian et al., 2024) introduces a DAG topology for reasoning among thousands of agents, while EvoMAC (Hu et al., 2024c) proposes a self-evolving collaboration framework for software development. These emphasize scaling collaboration, whereas our system focuses on *task-specialized decomposition and coordination* for audio-text validation.

AFlow (Zhang et al., 2024b) formalizes workflows as DAGs of LLM-invoking nodes, and Worf-Bench/WorfEval (Qiao et al., 2024) evaluates planning quality through subsequence and subgraph metrics. We adapt such methods to speech by grounding workflows in dataset curation and evaluating with domain-specific metrics such as word error rate (WER), character error rate (CER), and alignment accuracy.

AgentPrune (Zhang et al., 2024a) reduces communication redundancy, and T3-Agent (Gao et al., 2024b) improves multi-modal tool usage (Gao et al., 2024a). AutoAgent (Tang et al., 2025), AgentSquare (Shang et al., 2024), and automated design frameworks (Hu et al., 2024b) simplify agent creation. In contrast, our system emphasizes *robust reusable tools* for speech data (e.g., VAD, IndicLID (Madhani et al., 2023), CTC validators), integrated into an *end-to-end verification pipeline* that handles heterogeneous formats, multilingual scripts, and demographic balance.

Unlike NVIDIA Speech Explore[2], which is limited to transcript verification with 8 tasks, **SpeechQC-Agent** supports 24 overall verification tasks, 13 dedicated to transcripts, while also extending to audio and metadata level checks. It is the first agentic system applied to real-world speech corpus curation, combining workflow-graph metrics with speech-specific quality indicators.

## 3 DATA CREATION PIPELINE

We develop SpeechQC-Dataset, a synthetic speech-text benchmark designed to capture realistic quality control (QC) challenges. Unlike prior synthetic pipelines that primarily generate clean training data, the SpeechQC-Dataset is explicitly QC-aware: it integrates controlled noise, multilingual translation, and LLM-as-a-Judge validation to simulate diverse error conditions. This makes it the first dataset design to evaluate verification agents.

The general workflow for creating synthetic data is illustrated in Figure 2. The pipeline proceeds in five stages:

1. **Prompt-Driven Dialogue Generation**: A carefully crafted natural language prompt encodes the intention of the task, speaker roles, and domain constraints. An LLM-based dialogue planner (Deng et al., 2023; Yi et al., 2024) expands this into multi-turn conversations, guided by long-form examples for discourse coherence. A controller agent may invoke LLM-as-a-Judge (Gu et al., 2024b) to enforce factuality and conversational realism.

---

[1]**Code and Dataset Availability:** https://anonymous.4open.science/r/SpeechQC-Agent-B971/

[2]https://docs.nvidia.com/nemo-framework/user-guide/latest/nemotoolkit/tools/speech_data_explorer.html

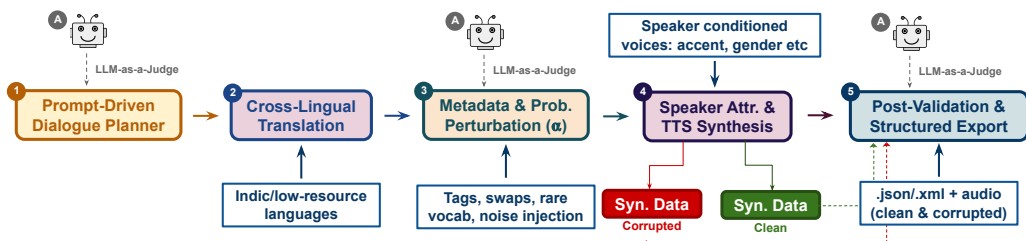

Figure 2: The five-stage data creation pipeline used for SpeechQC-Dataset: (1) prompt-driven dialogue planning, (2) cross-lingual translation, (3) metadata extraction and probabilistic perturbation, (4) speaker attribution & TTS synthesis, and (5) post-validation with structured export. LLM-as-a-Judge checkpoints (dashed) provide filtering and consistency checks at key stages.

2. **Cross-Lingual Translation**: The generated dialogues are translated into low-resource languages using multilingual LLMs (Joshi et al., 2024) or specialized translation modules, enabling coverage across Indic and other under-resourced settings.

3. **Metadata Extraction and Perturbation**: Conversations are automatically annotated with speaker turns, utterance boundaries, and intent labels. A probabilistic perturbation module introduces structured noise, such as tag insertions (<noise>, <html>), token swaps, or injection of rare vocabulary, under a tunable threshold parameter $\alpha$. This simulates systematic QC-relevant errors that are absent in previous corpora.

4. **Speaker Attribution and TTS Synthesis**: Synthetic speaker IDs condition TTS models to produce speech audio with varied voices, accents, prosody, and gender. This ensures demographic and acoustic diversity while maintaining alignment with transcripts.

5. **Post-Validation and Structured Export**: Both transcripts and audio undergo selective corruption (e.g., clipping, token drops) to mimic real-world ASR artifacts. The final outputs, text, metadata, and audio, are exported in standardized .json or .xml formats. In multiple stages, LLM-as-a-Judge (Gu et al., 2024a) is invoked to detect hallucinations or inconsistencies, filtering low-quality samples.

The resulting dataset provides fine-grained control over linguistic, acoustic, and structural properties, yielding a benchmark that is both scalable and realistic for evaluating multi-agent QC systems.[3]

## 3.1 Data Quality Verification

To systematically evaluate dataset quality, we design the SpeechQC-Dataset to include controlled errors injected through rule-based perturbations and LLM-as-a-Judge validation (Gu et al., 2024a). This allows us to benchmark verification methods against realistic error distributions.

The verification process is implemented as a two-stage suite that mirrors the multi-agent architecture of SpeechQC-Agent: (i) **QC1** for audio and metadata validation, and (ii) **QC2** for transcript and content validation. This modular separation enables parallelization while covering complementary error types. For further details, see Appendix E.

**Checklist Overview**: Table 1 summarizes all the QC1 and QC2 checks. Together, they capture a broad spectrum of quality dimensions, from file integrity and speaker uniqueness to transcription reliability, script consistency, and domain balance. Unlike prior ad hoc QC efforts, this structured design provides the first unified, evaluation-aware benchmark for speech dataset verification.

## 4 Methodology

In this section, we describe our proposed SpeechQC-Agent, a natural language-driven, LLM-coordinated multi-agent framework designed specifically for speech dataset quality verification. The framework takes as input a batch of speech data (waveforms, transcripts, metadata) and a natural-

---

[3]For further details, please refer to Appendix D.

| QC1: Audio & Metadata Verification (11 tasks) | | | |
|---|---|---|---|
| ASR Transcription | Speaker count & duration | Audio length | Silence calculation (VAD) |
| Upsampling check | Speaker identity (new vs. old) | Audio corruption check | Audio format / extension check |
| Sample rate check | Speaker duration stats | Language ID (ASR + IndicLID) | |
| QC2: Transcript & Content Verification (13 tasks) | | | |
| Transcript quality | Grapheme / character stats | Vocabulary stats | Language verification |
| English word counter | CTC score | Domain classification | Transcript-audio alignment |
| Transcript normalization (tags removal) | Transcript coherence (LLM-as-a-Judge) | Transliteration (Roman→Native) | Utterance duplicate check / WER computation |

Table 1: Overview of the 24 verification tasks implemented in SpeechQC-Agent, grouped into QC1 (audio/metadata) and QC2 (transcript/content).

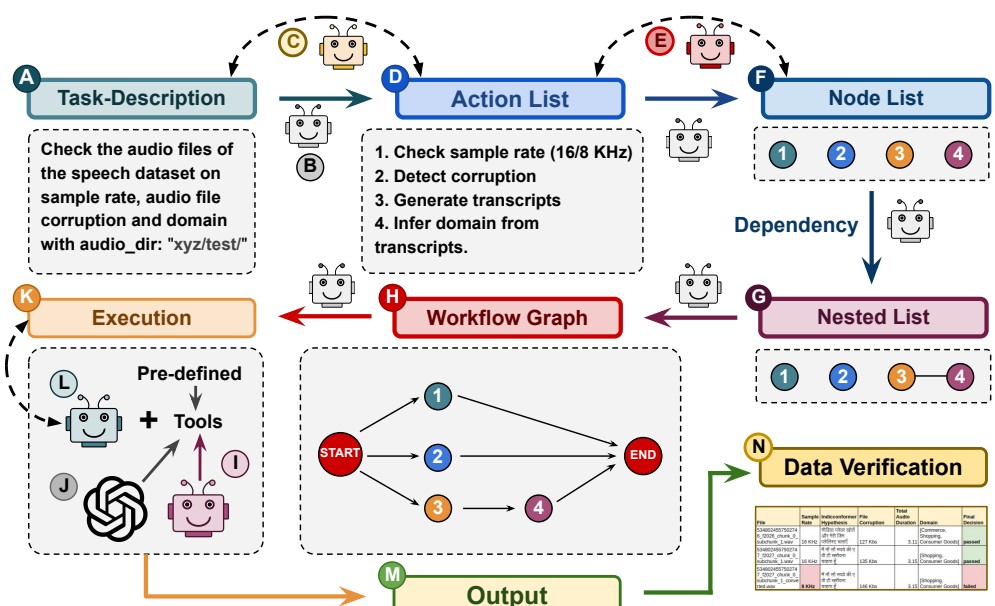

Figure 3: Architecture of the **SpeechQC-Agent** system. Given a natural language task description (A), a central planner LLM interprets the input and generates an ordered action list (C), which is validated (D) and mapped to a node list (E). Dependencies among nodes are resolved into a nested structure (F) and used to generate a topologically sorted workflow graph (G). Each node is executed (H) using dynamically synthesized (I) or pre-defined tools (J). Execution is monitored for completeness, and outputs (L) are aggregated into a structured Data Verification Dashboard (M) for final review.

language verification request. It then (i) decomposes the request into atomic checks, (ii) builds an executable directed acyclic graph (DAG) of those checks, (iii) instantiates or retrieves the required tools, and (iv) executes the graph while monitoring progress. Each stage in SpeechQC-Agent is powered by a specialized agent with its own planning, execution, or verification role, and these agents are coordinated by a central LLM planner (Wei et al., 2025). Figure 3 illustrates the pipeline, which consists of the following stages:

## 4.1 TASK PARSING AND ACTION GENERATION

The SpeechQC-Agent: central planning agent (A-B) is the primary interface for interacting with the user. It receives tasks from the user, comprehends the tasks in natural language tasks description $q$, a central planning agent leverages an LLM to interpret and decompose it into a structured action list $A = \{a_i\}_{i=1}^n$, where each action $a_i$ denotes a specific atomic quality check. Each action corresponds to a specific atomic quality check, such as CheckSampleRate , DetectAudioCorruption , or

ValidateTranscript etc. The PlannerLLM is a constrained sequence-to-sequence model that maps natural language queries to validated action lists.

$$A \leftarrow PlannerLLM_{\theta}(q)$$

This planning process uses a combination of lexical mapping (e.g., "VAD" $\rightarrow$ SilenceCheck ) and semantic prompting of an LLM to infer implied actions. Moreover, a lightweight rule-based LLM verifier ensures the consistency of $A$ with known hard constraints. For instance, if $q$ mentions silence or VAD, the agent appends $a_{silence}$ to $A$ if missing by the central planning agent.

## 4.2 Node Generation and Dependency Graph Construction

We define an agentic workflow as a series of LLM invokes in which the action list is further transformed into executable nodes $V = \{v_1, v_2, \ldots, v_n\}$. Each node $v_i$ represents a specific discrete verification subtask performed by an LLM. The dependencies between nodes are explicitly captured to construct a Directed Acyclic Graph (DAG) that is a pair $G = (V, E)$, where:

- $V$ is a finite set of vertices,
- $E \subseteq V \times V$ is a set of directed edges,

such that there does not exist any sequence of distinct vertices $v_1, v_2, \ldots, v_k \in V$ with $k \geq 2$ satisfying $(v_1, v_2) \in E$, $(v_2, v_3) \in E$, $\ldots$, $(v_{k-1}, v_k) \in E$, and $(v_k, v_1) \in E$ where edges $E$ represent dependencies between subtasks, which also govern the execution sequence.

Although graph structures can represent workflow relationships $W$, they require complex extensions beyond basic DAGs to naturally express parallel execution and conditional logic (Hu et al., 2024a). Neural networks enable adaptive transitions but lack precise control over workflow execution (Liu et al., 2023). In contrast, code representation inherently supports all the above relationships through standard programming constructs. Therefore, we adopt the code (Zhuge et al., 2024) as our primary edge structure to maximize expressivity. Then, the nodes are first linearly sequenced using topological sorting, where a linear ordering of the vertices is established such that each vertex appears before all vertices to which it has outgoing edges, followed by the establishment of parallel or sequential execution relationships:

$$C(V) \Rightarrow TopologicalSort \Rightarrow G$$

## 4.3 Tool Synthesis and Retrieval

Each verification node $v_i$ is associated with an executable tool $T_i$. Tools are selected or synthesized via: 1) Dynamic LLM-based tool synthesis $T_{gen}$: The agent prompts an LLM to generate the tools and callable functions. The tools are generated on demand for new or customized tasks. and 2) Predefined tool repository $T_{lib}$: A curated set of robust tools pre-generated using GPT-4o for stable performance when synthesis fails or confidence is low. The overall tool set is represented as:

$$T = T_{gen} \cup T_{lib}$$

The tool selection may be revised if the tool fails validation checks or runtime execution.

## 4.4 Workflow Execution and Monitoring

We execute the workflow graph $G$ following the topological order with dependency-aware parallelism. A monitoring agent uses a separate LLM to track the execution status of each node, ensuring completeness and robustness. Let $y^i$ be the output of node $v_i$. If $y^i = \emptyset$ or an exception is detected, the execution checker retries $T_i$ up to $r$ times. This guarantees completeness:

$$\forall v_i \in V, \quad \exists \hat{y}_i \neq \emptyset \vee \text{fail}(v_i)$$

Nodes that fail to execute are automatically retried or escalated for manual inspection[4].

---

[4]See Appendix K for all the prompts.

### 4.5 OUTPUT AGGREGATION AND DASHBOARD GENERATION

Upon completion, outputs from all executed nodes are aggregated into structured reports and visual dashboards. These dashboards enable users to interactively inspect and review quality metrics, error distributions, and execution logs for transparency and auditability. This enables both human analysts and downstream systems to filter or prioritize batches for the data verification task.

### 4.6 MODULARITY AND EXTENSIBILITY

SpeechQC-Agent is modular by design, with each stage (action parsing, node building, tool generation, execution) being LLM-agent pluggable. New tasks can be added by either extending the action ontology or defining a new node schema, supplying new tool definitions, or enabling auto-synthesis. This architecture generalizes across vendor schemas, speech domains, and languages without manual scripting, making it especially suited for multi-source, low-resource datasets. It also ensures adaptability and scalability to diverse speech dataset curation challenges.

## 5 EXPERIMENTS

### 5.1 DATASET CONSTRUCTION

To rigorously evaluate SpeechQC-Agent, we introduce the **SpeechQC-Dataset**, a synthetic yet QC-aware benchmark specifically designed for multilingual low-resource settings, with a focus on Indic languages, starting with Hindi language. Unlike prior synthetic corpora that emphasize clean training data, the SpeechQC-Dataset explicitly incorporates controlled variability and error patterns to stress-test verification systems.

The dataset is generated through a multi-step LLM+TTS pipeline (Section 3). Three strong LLMs: *LLaMA-3.3-70B*, *GPT-4o/4o-mini*, and *DeepSeek-R1-distill*, produce English multi-turn dialogues across 11 domains and 55 scenarios, ensuring conversational and stylistic diversity. The dialogues are translated into Hindi (Devanagari script) using multilingual LLMs, normalized, and converted into audio with a speaker-conditioned TTS model, yielding natural variation in voice, accent, and prosody.

Each sample is annotated with metadata fields including verbatim text, audio duration, scenario, domain, speaker ID, native language, gender, and age group. Speaker IDs and demographics are adapted from the LAHAJA dataset (Javed et al., 2024a), ensuring realistic population balance. Importantly, original Lahaja transcripts/audio are replaced with synthetic content, preserving demographics while avoiding data leakage. For further details, see Appendix C.

### 5.2 QUALITY VERIFICATION FRAMEWORK

We benchmark SpeechQC-Agent on a two-stage verification suite aligned with its modular architecture: **QC1: Audio & Metadata Verification** - Includes language ID, file format integrity, sample rate, silence/noise, upsampling artifacts, speaker reuse detection, and speaker-hour balance. **QC2: Transcript & Content Verification** - Includes forced alignment, CTC loss (wav2vec2.0), ensemble WER/CER scoring, LLM-as-a-Judge fluency scoring, domain classification, transliteration consistency, grapheme distribution, vocabulary rarity and duplication detection.

Table 1 enumerates all 24 checked implementations. To control difficulty, the dataset is partitioned into subfolders: QC1/2-1 (single transformation), QC1/2-2 (randomly paired transformations), and QC1/2-3 (three or more transformations). This design allows for controlled evaluation verification under increasing error complexity.

### 5.3 BASELINES AND LLM VARIANTS

We evaluated SpeechQC-Agent with five planner LLMs: *GPT-4o*, *GPT-4.1*, *DeepSeek-R1-distill*, *LLaMA-3.3-70B*, and *LLaMA-3.1-8B*.
1. **Execution Accuracy**: proportion of verification actions producing correct results given ground-truth annotations.

2. **Hallucination Rate**: fraction of actions not present in the requested query but introduced by the planner.

3. **Runtime**: average end-to-end wall-clock time per verification batch.

4. **Cost Efficiency**: normalized token cost per 1K tokens (see Appendix B.1).

We compare against one non-agentic and two agentic baselines: (i) a **Human Annotation baseline**, simulated through 5 annotators performing random 10% sampling. These establish lower/upper bounds for scalability and quality. (ii) a **Single-Agent** parses the user query, creates an action plan, and directly executes all verification tasks sequentially without explicit decomposition or dependency modeling, and (iii) a **Multi-Agent** handle subtasks in parallel, but without a central LLM planner or dependency graph, leading to limited coordination and potential ordering errors across tasks.

Our evaluation addresses three questions:

(1) Can the SpeechQC-Agent generalize verification workflows across languages and vendor formats?

(2) Which planner LLM offers the best trade-off between accuracy, hallucination, and cost?

(3) How does agentic verification compare against static pipelines and human-in-the-loop sampling?

| Model | File-Format | Corrupt | Sample-Rate | Domain |
|---|---|---|---|---|
| gpt-4o-mini | 100 | 100 | 100 | 28.17 |
| gpt-4.1-mini | 100 | 100 | 100 | 60.32 |
| deepseek-r1 | 0 | 100 | 0 | 81.30 |
| llama-3.1-8b | 0 | 100 | 0 | 21.48 |
| llama-3.3-70b | 100 | 100 | 100 | 75.40 |

Table 2: QC1 evaluation across four subtasks. Detect file format error, corrupt file error, sample rate error, and domain identification error. Reported values correspond to accuracy.

| LLM Variant | Roman Script | Mean WER | # HTML Tags | # EN Tokens |
|---|---|---|---|---|
| gpt-4.1-mini | 99.64 | 0.094 | 99.96 | 100 |
| gpt-4o-mini | 66.40 | 1.151 | 98.33 | 68.17 |
| deepseek-r1 | 85.2 | 0.334 | 0 | 0 |
| llama-3.1-8b | 0 | 0 | 0 | 91.34 |
| llama-3.3-70b | 99.46 | 0.120 | 100 | 78.70 |

Table 3: QC2 evaluation across four subtasks. Lower WER, Roman script, HTML Tags, and English Tokens, the accuracy calculated based on detection.

| Model | Accuracy (%) | Time (sec) |
|---|---|---|
| gpt-4o-mini | 75 | 354.207 |
| gpt-4.1-mini | 75 | 136.545 |
| llama-3.3-70b | 62.5 | 64.056 |
| llama-3.1-8b | 25 | 80.326 |
| deepseek-r1 | 25 | 72.693 |

Table 4: LLM performance on QC1 instructions. Each model was evaluated using 3 audio samples each from QC1-1, QC1-2, and QC1-3. The number of missed tasks and total execution time (in seconds) are reported.

## 6 RESULTS AND DISCUSSION

Tables 2 and 3 show that SpeechQC-Agent consistently performs verification tasks across heterogeneous data sources. In QC1 (audio and metadata), all strong planner LLMs achieve near-perfect detection of file-format errors, corruption, and sample rate anomalies, confirming robustness to vendor-side audio inconsistencies. In QC2 (transcripts and content), clearer differences emerge: GPT-4.1-mini and LLaMA-3.3-70B both surpass 99% accuracy in Roman script and HTML tag detection with low WER scores, while smaller models degrade substantially. Importantly, SpeechQC-Agent transfers effectively from Hindi Devanagari-script benchmarks to additional Indic languages

| Model | Accuracy | Hallucinated Tasks | Time (sec) |
|---|---|---|---|
| gpt-4o-mini | 100 | 0 | 347.397 |
| gpt-4.1-mini | 100 | 0 | 298.261 |
| deepseek-r1 | 12.5 | 2 | 65.314 |
| llama-3.1-8b | 33 | 0 | 60.542 |
| llama-3.3-70b | 72 | 0 | 37.380 |

Table 5: Evaluation of LLMs on QC2 instructions. Each model was prompted to execute 8 quality verification tasks (language check, WER/CTC computation, normalization, etc.) for 3 audio samples. Models were evaluated based on task completion accuracy and hallucination rate.

in our extended in-house corpus, suggesting that its workflow abstraction generalizes beyond the synthetic dataset. Together, these results validate our claim that SpeechQC-Agent can handle both cross-lingual variability and vendor-specific data formats.

Tables 4 and 5 compare planner LLMs on execution accuracy, hallucination rate, and runtime. GPT-4.1-mini achieves the strongest transcript-level fidelity (99.64% in Roman script, 0.094 mean WER) with zero hallucinated steps, albeit at a moderate runtime cost (136-298s per batch). LLaMA-3.3-70B offers the best trade-off between efficiency and accuracy, completing QC pipelines in under 40s with only minor accuracy degradation. By contrast, smaller models such as GPT-4o-mini struggle with high hallucination rates, while DeepSeek-R1, although less reliable for QC pipelines, proves highly effective for reasoning-oriented tasks such as domain classification. Overall, the results reveal a flexible deployment spectrum: high-stakes verification may prioritize GPT-4.1-mini for maximal fidelity, LLaMA-3.3-70B enables cost- and time-efficient batch verification, and DeepSeek-R1 supports logical checks like domain identification. We also used SpeechQC-Agent to check large-scale real vendor corpora and to determine the domain of the IndicVoice (Javed et al., 2024b) dataset (see Appendix H.1 and H.2).

The results collectively underscore three findings. First, the QC-aware dataset design, incorporating controlled variability and perturbations, enables robust generalization across languages and vendor formats. Second, modular multi-agent orchestration ensures that planner LLMs can be swapped to balance accuracy and efficiency, with GPT-4.1-mini excelling in reliability and LLaMA-3.3-70B in speed. Finally, SpeechQC-Agent achieves 80-90% of expert-level verification accuracy while reducing runtime and cost to less than 20% expert annotation (see Appendix H.3). Taken together, these findings validate SpeechQC-Agent as the first scalable end-to-end framework for the verification of the quality of the speech dataset in low-resource multilingual settings[5]. We include a detailed case study in the Appendix F to demonstrate the generalizability of SpeechQC-Agent to real-world scenarios.

## 7 CONCLUSION

We introduced **SpeechQC-Agent**, the first natural language-driven multi-agent framework for scalable speech dataset quality verification. Unlike static rule-based pipelines or human-in-the-loop sampling, our approach leverages a central planner LLM to decompose user queries into DAG-based workflows that seamlessly integrate reusable and LLM-synthesized tools. Our evaluations show that **LLaMA-3.3-70B** and **DeepSeek-R1** together provide an effective balance, handling large-scale verification and logical tasks such as domain classification while maintaining cost and time efficiency without sacrificing accuracy. Alongside the release of SpeechQC-Dataset, a realistic multilingual benchmark, we demonstrate that SpeechQC-Agent achieves 80-90% of expert-level verification accuracy in less than 20% of cost and time, offering a flexible and practical solution for large-scale speech corpus curation.

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

## A  APPENDIX

In the Appendix, we provide:

1. Section B: Compute Infrastructure
2. Section C: Dataset Composition
3. Section D: SpeechQC-Dataset
4. Section E: Data Quality Verification Framework
5. Section F: Case Study Example
6. Section G: Tasks Information
7. Section H: Further Analysis
8. Section I: Future Directions
9. Section J: Limitations
10. Section K: Prompts

## B  COMPUTE INFRASTRUCTURE

**Compute details**: For all our pre-training and fine-tuning experiments, we used two NVIDIA A100-SXM4-80GB GPUs. Each training requires 4-48 hours.
**Software and Packages details**: We implement all our models in PyTorch[6].

### B.1  INFERENCE COST

Table 6 compares the inference costs across LLMs, showing that open-source models like LLaMA-3.3-70B and LLaMA-3.1-8B are substantially cheaper than GPT variants, making them attractive for large-scale deployment.

| Model | Cost |
|---|---|
| gpt-4o-mini | $0.06 |
| gpt-4.1-mini | $0.08 |
| llama-3.3-70b | < $0.03 |
| llama-3.1-8b | $0.02 |
| deepseek-r1 | $0.05 |

Table 6: Inference cost (USD / 1K tokens) per 1,000 tokens for different LLMs.

## C  DATASET COMPOSITION

To evaluate the robustness of our audio and transcript quality control mechanisms, we constructed a synthetic dataset with intentional flaws from the LAHAJA dataset and custom-generated data. The dataset comprises four subsets to test specific quality control aspects across diverse error profiles and sources.

- **Vendor A (Audio-Specific):** Applied QC1 transformations (e.g., File Format Conversion, Corrupt File Simulation, Sample Rate Reduction) to 3,000 LAHAJA entries (1,000 individual, 1,000 paired, 1,000 multiple QC1).
- **Vendor B (Transcript Quality):** Applied QC2 transformations (e.g., Audio-Transcript Misalignment, Script Inconsistency, Transcript De-normalization) to 3,000 LAHAJA entries (1,000 individual, 1,000 paired, 1,000 multiple QC2).
- **Vendor C (Mixed Flaws):** Applied both QC1 and QC2 transformations to 100 random LAHAJA entries for combined audio-transcript testing.
- **Vendor D (Synthetic Data):** Generated an independent dataset using LLMs and TTS models for synthetic audio and transcripts with controlled quality parameters.

---

[6]https://pytorch.org/

## C.1 ADDITIONAL DATA INFORMATION

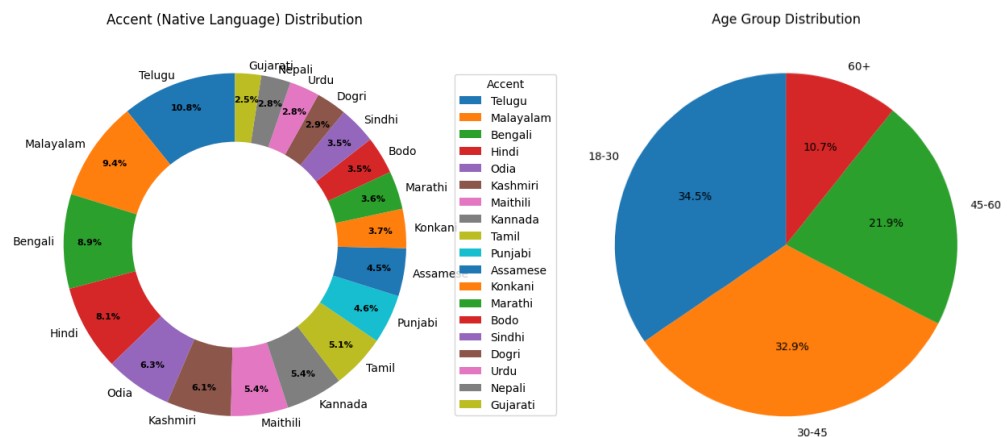

Figure 4: Distribution of speakers in the **SpeechQC-Dataset** by native language accent (left) and age group (right). The dataset exhibits broad linguistic diversity, with representation from 19 native languages, and covers a wide range of age groups, ensuring demographic balance for robust speech technology evaluation.

It includes 15.51 hours of Hindi speech data from 110 unique speakers, with a balanced gender split of 54 female and 56 male. Speakers cover age groups of 18-30, 30-45, 45-60, and 60+, and represent 19 native languages, led by Telugu, Malayalam, Bengali, Hindi, and others (Fig 4). The data set spans 11 domains, such as agriculture and science and technology, in 55 conversational settings (Table 7). Its strength lies in its extensive demographic and linguistic diversity, paired with broad domain coverage, making it a vital tool for inclusive speech technologies.

# D SPEECHQC-DATASET

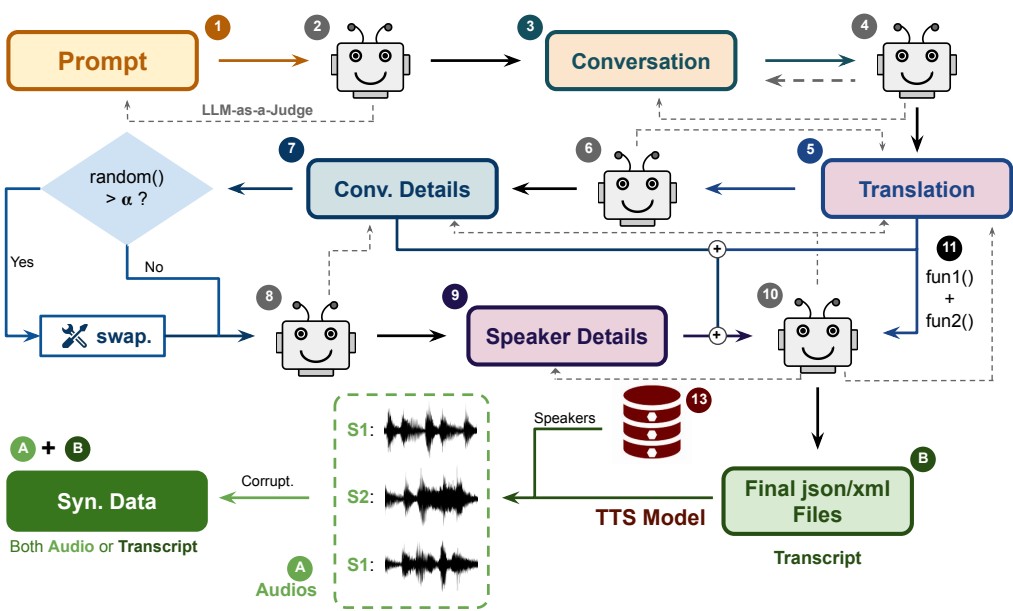

Figure 5: SpeechQC-Dataset generation pipeline. Each numbered step corresponds to an LLM or tool-based operation within the multi-LLM workflow.

| Domain | Setting | LLM Used |
|---|---|---|
| Indian Agri. | Village farm on crops | GPT Models |
| | Agri fair innovations | GPT Models |
| | Rural sustainable workshop | Llama Model |
| | Farmers' crop tips | Llama Model |
| | School farm trip | DeepSeek |
| Indian Law | Mock court basics | GPT Models |
| | Library governance talk | GPT Models |
| | Civic rights discussion | Llama Model |
| | Town hall governance | Llama Model |
| | Constitution lecture | DeepSeek |
| Indian Finance | Budgeting workshop | GPT Models |
| | Digital banking expo | GPT Models |
| | Savings community chat | Llama Model |
| | Loan process at bank | Llama Model |
| | Banks' role in class | DeepSeek |
| Indian Sports | Sports event at park | GPT Models |
| | Movie event planning | GPT Models |
| | Fitness benefits in gym | Llama Model |
| | Dance prep in area | Llama Model |
| | Cinema fan club | DeepSeek |
| Indian Military | Fitness drills camp | GPT Models |
| | Military history talk | GPT Models |
| | Veterans' community event | Llama Model |
| | Defence awareness seminar | Llama Model |
| | Armed forces career fair | DeepSeek |
| Indian Politics | Democracy school talk | GPT Models |
| | Political history session | GPT Models |
| | Civic duties debate | Llama Model |
| | Voting cultural event | Llama Model |
| | Civic podcast | DeepSeek |
| Indian Edu. | Rural learning school | GPT Models |
| | Student science fair | GPT Models |
| | Exam study group | Llama Model |
| | Parent-teacher engagement | Llama Model |
| | University education day | DeepSeek |
| Indian Science | Tech innovation exhibit | GPT Models |
| | Basic coding workshop | GPT Models |
| | Tech future school club | Llama Model |
| | Eco-tech startup hub | Llama Model |
| | Digital tools outreach | DeepSeek |
| Indian Rural Dev. | Infrastructure village meet | GPT Models |
| | Sanitation campaign | GPT Models |
| | Amenities workshop | Llama Model |
| | Renewable energy event | Llama Model |
| | Model village project | DeepSeek |
| Indian Business | Entrepreneurship fair | GPT Models |
| | Small business seminar | GPT Models |
| | Trade at marketplace | Llama Model |
| | Supply-demand class | Llama Model |
| | Financial planning | DeepSeek |
| Indian Art | Art evolution exhibit | GPT Models |
| | Cultural fair performance | GPT Models |
| | Modern art club | Llama Model |
| | Architecture history | Llama Model |
| | Heritage preservation | DeepSeek |

Table 7: Domains and Settings with LLM Attribution

We develop SpeechQC-Dataset, a synthetic dataset generation framework powered by multi-LLMs. As illustrated in Figure 5, the pipeline simulates realistic conversational interactions, diverse speaker characteristics, and common ASR artifacts, producing structured audio-text pairs annotated with rich metadata. Unlike prior synthetic pipelines focused on text-only generation, SpeechQC-Dataset integrates dialogue planning, cross-lingual translation, perturbation, and TTS-driven synthesis into a unified workflow tailored for speech quality verification.

1. **Prompt Initialization**: The pipeline begins with a carefully designed prompt (Figure 5, Step 1) encoding task-specific intent, speaker roles, or domain constraints. This is passed to an LLM agent that orchestrates conversation planning.

2. **Conversation Generation**: An LLM agent (Step 3) generates multi-turn dialogue from the prompt, simulating realistic human interactions. Long-form in-context examples improve coherence and pragmatic diversity. A controller agent (Step 2) may invoke an LLM-as-a-Judge to monitor factuality and conversational realism.

3. **Translation**: Conversations are translated into one or more low-resource languages using multilingual LLMs or specialized modules (Step 5), ensuring coverage and supporting cross-lingual generalization.

4. **Metadata Extraction**: A parsing module extracts fine-grained metadata, including turn segmentation, utterance boundaries, intent types, and context tags (Steps 6-7).

5. **Probabilistic Perturbation**: Controlled variability is introduced through randomized operations such as tag insertion (e.g., `<noise>`, `<html>`), token swaps, or word-level noise (Step 8). The perturbation is applied according to a threshold parameter $\alpha$ which simulates the variation of the structured data.

6. **Speaker Attribution**: Synthetic speaker IDs are assigned to utterances (Step 9) to simulate multi-speaker settings. These annotations condition downstream TTS synthesis, enabling diversity in voice, accent, prosody, and gender.

7. **TTS-driven Audio Synthesis**: A TTS model synthesizes audio from transcripts and speaker metadata (Step 13). Speaker-conditioned synthesis ensures demographic and prosodic diversity.

8. **Optional corruption**: To mimic real-world ASR challenges, both audio and transcripts may be selectively corrupted (e.g., dropped tokens, clipping). These corrupted variants support robustness evaluation (left branch of "Syn. Data").

9. **Structured Conversion**: The final outputs, including transcripts, metadata, and audio files, are exported in standardized formats (`.json`, `.xml`) for compatibility with the downstream QC tasks (Step B).

10. **Post-Processing and Validation**: Custom functions (e.g., `fun1()`, `fun2()`, Step 11) perform final consistency checks and metadata linking. An LLM-as-a-Judge (Gu et al., 2024a) may be invoked at multiple stages (Steps 2, 4, 6, 10) to detect missing information or hallucinations.

Overall, this pipeline provides fine-grained control over dataset characteristics while leveraging LLM-based creativity, yielding a benchmark tailored for evaluating and training multi-agent speech verification systems.

## E  DATA QUALITY VERIFICATION FRAMEWORK

Figure 6 illustrates the two-stage data quality verification framework implemented in SpeechQC-Agent. **QC1: Audio and Metadata Verification.** As shown in the left panel of Figure 6, QC1 begins by applying a multilingual language identification model to determine the spoken language directly from the audio (Step 1). Subsequent checks validate the audio format, sampling rate, silence duration, frequency upsampling artifacts, and number of channels (Step 2). To handle speaker-related verification, we use speaker embedding-based clustering to identify unique speakers (Step 4) and validate whether speakers are reused across batches by comparing against known public and private datasets (Step 3). Additional statistics, such as the number of speakers (Step 5) and total speaking time per speaker (Step 6), are computed to assess speaker diversity and duration balance.

Figure 6: Overview of the SpeechQC-Agent quality control framework. QC1 (left) verifies audio and metadata through checks such as language ID, format integrity, silence detection, and speaker diversity. QC2 (right) validates transcripts using alignment, CTC loss, WER/CER ensembles, LLM-based fluency scoring, domain classification, and duplication detection. Together, these two stages provide comprehensive coverage of audio- and text-level dataset quality.

**QC2: Transcript and Content Verification.** The right panel of Figure 6 depicts QC2, which starts by aligning the transcript with the corresponding audio using a timestamp-agnostic model (Step 1). Transcript quality is scored using three different metrics: (i) Connectionist Temporal Classification (CTC) loss from a pretrained wav2vec2.0 model, (ii) Mixture-of-Experts (MoE) relative WER/CER scores from multiple ASR models, and (iii) LLM-as-a-Judge scores evaluating fluency and coherence (Step 2). Domain labels (Step 3) are inferred to ensure topic coverage and diversity. Further checks analyze the distribution of graphemes (Characters) and vocabulary rarity (Step 4), using an internal vocabulary list (Step 5). Finally, content duplication is measured using n-gram and embedding-based overlap with both intra-dataset and public corpus references (Step 6).

## F   CASE STUDY EXAMPLE

To illustrate the functionality of SpeechQC-Agent, we present a concrete example (Figure 7). The user submits the following natural language query:

> *"Check the audio files of the speech dataset on sample rate, audio file corruption, and domain with audio_dir: xyz/test/."*

The Orchestrator Agent parses this query and decomposes it into four atomic verification tasks: (1) validate the audio sampling rate, (2) detect corrupted files, (3) generate transcripts, and (4) infer domain labels from transcripts.

A Prompt Checker ensures the task list is complete and consistent (e.g., enforcing that transcript generation precedes domain inference). It re-verifies selected tasks and iteratively patches missing ones (up to three iterations) before graph build. The validated list is then compiled into a directed acyclic graph (DAG), where dependencies are explicitly encoded. Topological sorting guarantees that independent checks (e.g., sample rate and corruption detection) run in parallel, while dependent tasks follow sequential order. The Constrained topological scheduling and node-level validators are used in the pipeline to prevent hallucination (Task Selection / Topological Sorting / Prompt Checker prompts in Appx. K) .

Each DAG node is mapped to an execution agent:
- **Predefined Agent Nodes** (e.g., sample_rate_agent) handle standard checks such as verifying whether each file is at 16 kHz or 8 kHz.
- **Python REPL Tool** is invoked for low-level operations such as scanning directories or reading audio headers.
- **Predefined Tools** (e.g., silence detectors, WER scorers) are reused for stability.
Intermediate outputs are stored in a shared **State Dictionary**, ensuring later tasks (e.g., domain inference) can access transcripts generated earlier.

The final results are compiled into structured CSV files. For example, the sample-rate check produces a file with entries of the form:

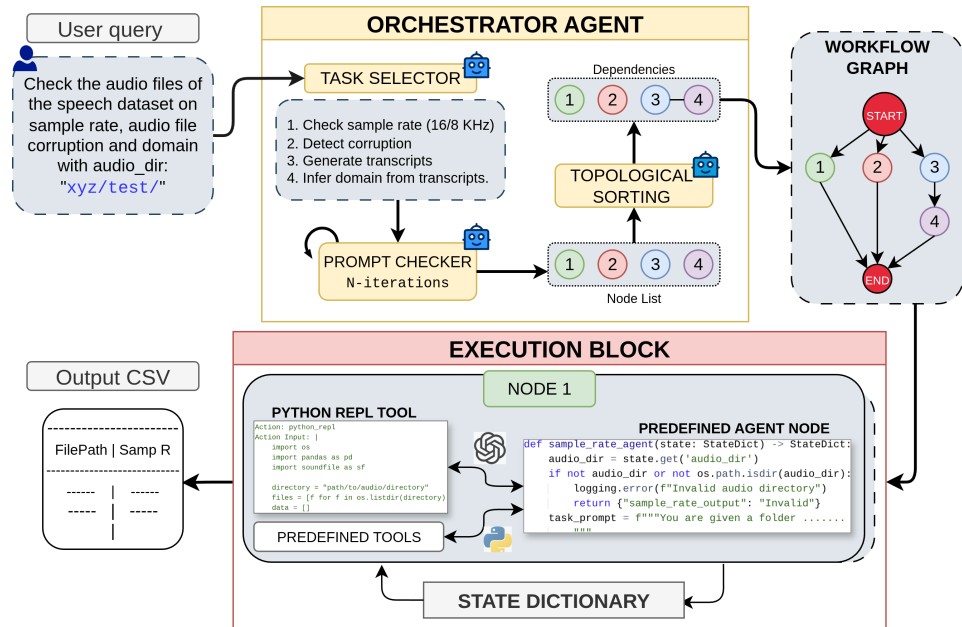

Figure 7: Illustrative example of SpeechQC-Agent executing a user query. The user requests checks on sample rate, file corruption, and domain inference for an input directory. The Orchestrator Agent decomposes the query into atomic tasks, validates them, and constructs a DAG workflow via topological sorting. Each node is executed within the Execution Block using either predefined agent nodes, dynamic Python REPL tools, or reusable predefined modules. Intermediate results are stored in a shared state dictionary and aggregated into structured outputs (e.g., CSV reports). This case study highlights the system's ability to translate natural language instructions into end-to-end, dependency-aware verification pipelines.

```
FilePath          | Samp R
------------------|--------
xyz/test/a1.wav   | 16000
xyz/test/a2.wav   | Invalid
```

This case study highlights how SpeechQC-Agent automatically translates a single user query into a multi-step verification workflow as shown in Figure 8. The system dynamically integrates LLM-synthesized tools with reusable predefined modules, respects dependencies through DAG orchestration, and outputs human-interpretable reports. This functionality demonstrates the practicality and flexibility of our agentic design for large-scale speech dataset verification.

## G  TASKS INFORMATION

SpeechQC-Agent implements a comprehensive suite of verification modules that span both audio (QC1) and transcript/content (QC2) modalities, enabling fine-grained inspection of speech datasets (Table 8).

Together, QC1 and QC2 provide 24 modular checks covering file integrity, speaker diversity, linguistic correctness, and content variety. This dual-stage design enables SpeechQC-Agent to detect both low-level anomalies (e.g., corrupted audio, missing metadata) and high-level linguistic errors (e.g., script inconsistency, domain imbalance), making it suitable for benchmarking multilingual, low-resource speech datasets.

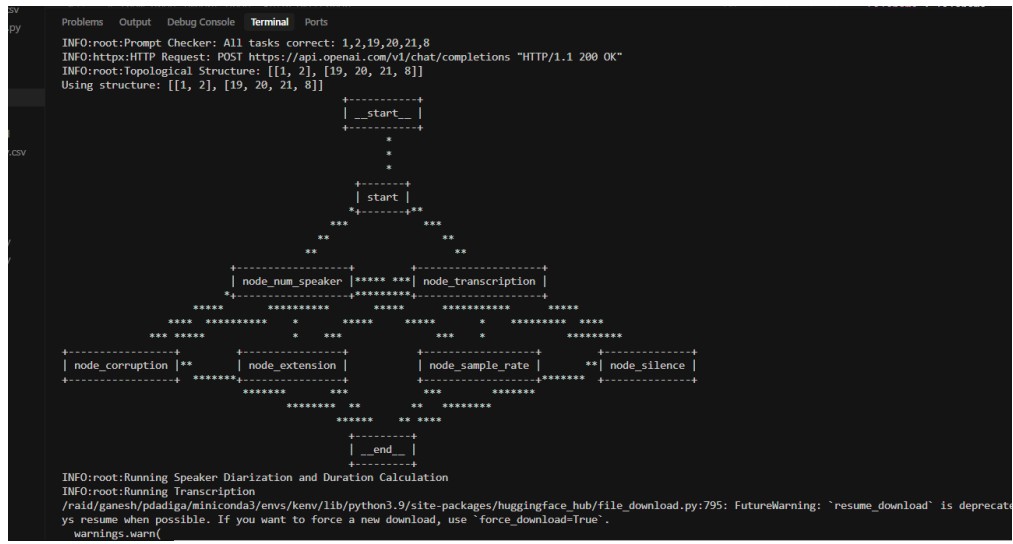

Figure 8: Complete audio processing summary for directory xyz/test/ including speaker diarization metrics, Voice Activity Detection (VAD) silence analysis, and comprehensive quality control validation

| QC Level | Check Name | Description |
|---|---|---|
| QC1 | Language Identification (Audio) | Uses multilingual models to detect the spoken language, independent of metadata. |
| QC1 | File Format | Validates that the audio file conforms to required encoding standards (e.g., WAV). |
| QC1 | Corrupt File Detection | Detects corrupted or zero-length audio files. |
| QC1 | Sample Rate Check | Verifies if the sample rate is 16 kHz and above. |
| QC1 | Silence Duration | Calculates the total silence duration per audio file. |
| QC1 | Upsampling Detection | Identifies audio upsampled from low fidelity (e.g., 8 kHz to 16 kHz). |
| QC1 | Number of Speakers | Estimates the number of unique speakers in the batch. |
| QC1 | Per-Speaker Duration | Measures cumulative speaking time per speaker to ensure speaker diversity. |
| QC1 | Speaker Validity | Checks if a speaker is repeated across batches (e.g., same vendor, same voice reused). |
| QC2 | Audio-Transcript Alignment | Aligns transcript with audio, regardless of format or file structure. |
| QC2 | Timestamp-Based Segmentation | Segments long audio using provided transcription time-stamps for utterance-level alignment. |
| QC2 | Script Consistency | Ensures transcript uses the correct native script, avoiding Romanized text unless intentional. |
| QC2 | Code-Mixing Detection | Identifies code-mixed utterances (e.g., English-Hindi), which may require for diverse dataset. |
| QC2 | MOE Score (Mixture of Expert) | Calculates WER and CER using multiple ASR models to quantify transcription quality. |
| QC2 | CTC Score | Computes the Connectionist Temporal Classification (CTC) loss using the wav2vec model. |
| QC2 | LLM Score | Evaluates transcript coherence and fluency using LLM-as-a-Judge. |
| QC2 | Transcript Normalization | Removes HTML tags, other tags, or other extraneous tokens from the transcript. |
| QC2 | Transliteration Consistency | Checks Roman-to-native script transliteration for consistent representation. |
| QC2 | Grapheme | Analyze the distribution of different characters. |
| QC2 | Vocabulary Coverage | Analyzes the distribution of rare words. |
| QC2 | Domain Classification | Assigns each sample to domain labels (e.g., agriculture, etc) to ensure topic diversity. |
| QC2 | Content Repetition Check | Flags duplicated or reused content within or across datasets, including public corpora overlaps. |

Table 8: Overview of Data Quality Control (QC) Modules used by the SpeechQC-Agent for analyzing SpeechQC-Dataset and other speech datasets.

# H  FURTHER ANALYSIS

Table 9 shows that while GPT-4.1-mini and LLaMA-3.3-70B maintain high accuracy across QC2 tasks, **LLaMA-3.1-8B performs poorly**, with accuracies dropping below 30% and frequent task failures. This weakness stems from two factors: (i) its smaller scale limits compositional reasoning over multi-step workflows, and (ii) unlike larger models, it has been trained on comparatively less (or no) Indic language data, making it ill-suited for tasks involving script consistency, transliteration, and code-mixing checks. As a result, the model struggles to interpret instructions and generate valid verification outputs, highlighting the importance of both model capacity and linguistic coverage for robust dataset quality control.

Table 10 presents a comparative analysis of LLM performance in QC1 tasks involving audio and metadata verification. Among all evaluated models, GPT-4o-mini exhibited the most reliable behavior, successfully completing all five tasks, including file format validation, corruption detection, sample rate checking, speaker duration estimation and speaker validity matching. GPT-4.1-mini

| Model | Task | Accuracy | Hallucination |
|---|---|---|---|
| **GPT-4o-mini** | QC2-1 | 92.34 | 0 |
| | QC2-2 | 91.49 | 0 |
| | QC2-3 | 47.99 | 0 |
| **gpt-4.1-mini** | QC2-1 | 100 | 57.21 |
| | QC2-2 | 100 | 10.89 |
| | QC2-3 | 98.07 | 6.35 |
| **llama-3.1-8b** | QC2-1 | 27.64 | 10.57 |
| | QC2-2 | 18.02 | 0 |
| | QC2-3 | 0 | 0 |
| **llama-3.3-70b** | QC2-1 | 100 | 0 |
| | QC2-2 | 95.74 | 0 |
| | QC2-3 | 91.58 | 0 |

Table 9: Evaluation of different LLMs on quality control tasks (QC2-1 to QC2-3) measuring Accuracy and Hallucination rate in percent.

| LLMs | File Format | Corrupt File | Sample Rate | Speaker (Duration) | Valid Speaker |
|---|---|---|---|---|---|
| `llama-3.1-8b` | NP | NP | NP | NP (2/3) | NP |
| `llama-3.3-70b` | C | C | C | C | C |
| `deepseek-r1` | NP | NP | NP | C | NP |
| `gpt-4.1-mini` | C | C | C | C | Failed |
| `gpt-4o-mini` | C | C | C | C | C |

Table 10: Performance of LLMs on QC1 verification tasks. Each model is evaluated on its ability to execute file format validation, corruption detection, sample rate checks, speaker duration analysis, and valid speaker matching. Remarks provide qualitative insights into model behavior during task execution. (NP - Not Performing, C - Completed)

also performed well in most categories but failed to correctly handle valid speaker identification. In contrast, LLaMA-3.3-70b completed all core tasks, but introduced unnecessary operations, indicating weaker task-grounding. In particular, LLaMA-3.1-8b, while the fastest model, failed to perform most tasks and struggled with topological reasoning and task mapping. `DeepSeek-R1` demonstrated partial success in speaker tasks, but did not engage with other checks and required more iterations. These results highlight the trade-offs between speed, instruction-following capability, and task reliability across model families, reinforcing the need for instruction-based evaluation in speech data quality workflows.

| Protocol Block | Metric | In Sheet? | Missing Evidence / Action |
|---|---|---|---|
| *QC1 - Audio & Metadata* | | | |
| File format & corruption | Accuracy | 2 | Log SoX/ffprobe checks per file |
| Silence hours | Precision/Recall | 2 | Duration histograms with silence detector |
| Upsampling (8→16 kHz) | Binary accuracy | 2 | FFT-based up-sampling flag per file |
| Language ID (MMS) | Accuracy | 2 | MMS predictions + meta-tags |
| Speaker hours / diversity | Completeness, SDI | 2 | Diarisation output, per-ID hours |
| Speaker reuse detection | Match-rate | 2 | Embedding match vs. public pools |
| *QC2 - Transcript & Content* | | | |
| Audio-text alignment | WER, CER | 1 | CER still missing |
| Segmentation by timestamps | Seg. accuracy | 2 | Gold vs. predicted boundaries |
| Script validity | Script-match % | 3 | Need total-token denominator |
| CTC quality score | Avg. CTC | 1 | - |
| LLM-as-Judge rating | 1-5 score, | 2 | Per-utt. ratings + agreement |
| Normalization noise | HTML-error rate | 1 | Tag counts → rate per K tokens |
| Transliteration match | Accuracy | 2 | IndicTrans vs. transcript tokens |
| Vocab / grapheme diversity | Diversity score | 2 | Entropy or TTR statistics |
| Domain verification | Domain-match | 3 | Need gold domain labels |
| Duplication detection | Dup. score | 3 | Embedding-similarity counts |

Table 11: Coverage of the full QC-metric suite. 1 = logged, 2 = partially logged, 3 = not present.

Table 11 reveals substantial metric-coverage gaps: none of the QC-1 audio-metadata checks (format integrity, silence detection, up-sampling, language ID, speaker diversity or reuse) are logged, and several critical QC-2 dimensions, segmentation accuracy, CER, LLM-as-a-judge agreement, transliteration accuracy, vocabulary diversity, domain match, and duplication detection, remain unpopulated or only partially captured. Closing these gaps will require integrating raw SoX/ffprobe diagnostics, diarisation statistics, IndicTrans comparisons, lexical-entropy measures and embedding-based duplication scores, enabling a truly end-to-end, metrics-complete evaluation pipeline for future batches.

## H.1 SPEECHQC-AGENT ON REAL-WORLD VENDOR DATASETS

We deployed SpeechQC-Agent on a 12,000-hour multi-vendor speech corpus spanning Hindi, Assamese, Marathi, and other 5 Indic languages. The system automatically flagged large-scale quality issues, including 800 hours of Hindi audio with substandard sampling rates, 100 hours of Assamese and 200 hours of Marathi with Roman-script transcripts, 200 hours of silence-only audio, 200 hours of proxy (synthetic) speakers, 100 hours of misaligned audio-transcript pairs, and 500 hours of duplicated content across vendors. It also identified 987 corrupted audio files, while validating the majority of data as clean (Table 12). The hours flagged under each type of error were requested again from the vendors as replacement datasets.

Crucially, SpeechQC-Agent did not only surface errors, but also highlighted vendors consistently delivering high-quality corpora, enabling more informed procurement decisions and vendor accountability. Table 13 presents aggregate results: across the 6,000 hours, the system flagged 1,950 hours (32.5%) as problematic and validated 4,050 hours (67.5%) as reliable. The remaining datasets are currently under validation. These results illustrate that SpeechQC-Agent can perform systematic, large-scale dataset audits, offering both coverage and cost-efficiency beyond static pipelines or manual sampling.

| Error Type | Hours / Files Affected | Vendor Batches |
|---|---|---|
| Low sample rate (Hindi) | 800 hrs | Hindi vendor (B1) |
| Mis-scripted transcripts (Assamese, Marathi) | 100 hrs Assamese + 200 hrs Marathi | Assamese (B1), Marathi (B2) |
| Silence-only audio | 200 hrs | Multi-vendor (12k hrs total) |
| Proxy speakers | 200 hrs | Proxy vendor batch |
| Misaligned audio-transcript pairs | 100 hrs | Vendor B2 |
| Duplicate audio-transcript (cross-vendor) | 500 hrs | Across vendors/languages |
| Corrupted audio files | 987 files | Batch B1 (Hindi) |

Table 12: Summary of errors flagged by SpeechQC-Agent on real vendor datasets.

| Category | Hours | Percentage |
|---|---|---|
| Flagged problematic data | 1,950 | 32.5% |
| Validated clean data | 4,050 | 67.5% |
| Total | 6,000 | 100% |

Table 13: Aggregate audit of multi-vendor corpora (12,000 hours) using SpeechQC-Agent.

## H.2 PERFORMANCE OF LLMS ON INDICVOICE DATASET

| Model | Zero-Shot | 7-Shot |
|---|---|---|
| gpt-4o-mini | 0.314 | 0.254 |
| gpt-4.1-mini | 0.360 | 0.340 |
| deepseek-r1 | **0.456** | **0.386** |
| llama-3.1-8b | 0.180 | 0.110 |
| llama-3.3-70b | 0.338 | 0.352 |

Table 14: Performance of different models on the IndicVoice validation subset (500 utterances) across zero-shot and 7-shot settings.

Table 14 presents the domain classification performance of different LLMs in zero-shot and few-shot (7-shot) settings on the IndicVoice validation subset (Javed et al., 2024b) of 500 utterances.

DeepSeek-R1 achieves the highest accuracy across all conditions, with 45.6 in domain generalization, 0.456 in zero-shot classification, and 0.386 in few-shot classification. Its superior performance can be attributed to its reasoning-centric design, which enables it to capture contextual cues more effectively and make consistent domain assignments. In contrast, smaller models such as LLaMA-3.1-8B-Instant demonstrate weaker adaptability to domain variability. These results highlight the importance of reasoning-oriented LLMs in improving domain classification robustness for multi-domain Indic speech datasets. Furthermore, the relatively low WER observed in this setting can be explained by the nature of the utterances. For example, when a speaker refers to a "vessel," the utterance could plausibly belong to either the Daily Life Conversation domain (household usage) or the Product Review domain (product evaluation). Since both interpretations are semantically close and share overlapping vocabulary, therefore, it is difficult to identify the domain correctly.

## H.3   AGENTIC VS. STATIC AND HUMAN-IN-THE-LOOP PIPELINES

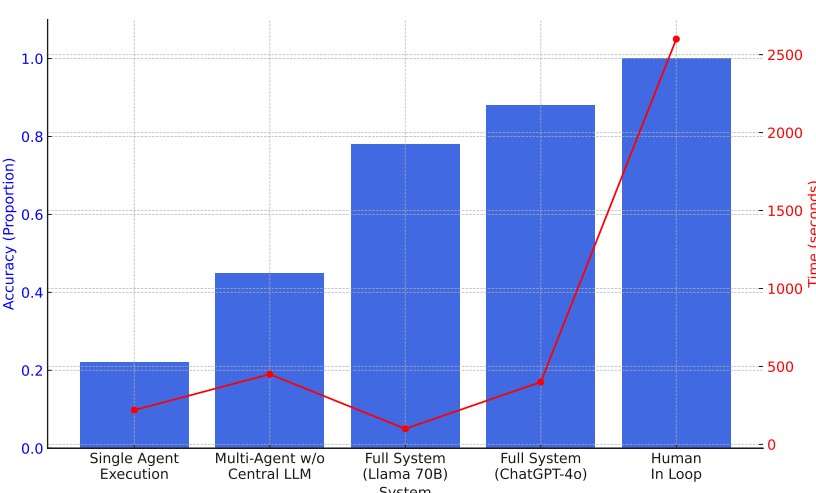

Figure 9: Comparison of verification accuracy (blue bars) and runtime (red line) across different system configurations. Static pipelines such as single-agent execution and no-central-LLM baselines achieve low accuracy despite faster execution. Full SpeechQC-Agent setups (with LLaMA-70B or GPT-4o as planner) substantially improve accuracy while keeping runtime manageable. Human-in-the-loop verification attains perfect accuracy but at the cost of prohibitively long runtimes.

Figure 9 contrasts SpeechQC-Agent with static pipelines and human verification. Static baselines (single-agent execution) achieve low accuracy ($\leq 0.4$) despite fast runtime, underscoring their inability to capture interdependent checks. In contrast, the full SpeechQC-Agent (with centralized LLM planning) attains accuracies above 0.8, approaching human-level reliability, while maintaining runtimes an order of magnitude lower than manual verification. The Human Verifier in this study is a speech technology expert, highly skilled in applying multiple techniques to assess the quality of speech datasets, and thus represents the gold standard ($\sim$1.0 accuracy). However, such expert verification is prohibitively expensive (more than 2500s per batch). SpeechQC-Agent therefore bridges the gap, providing 80-90% of expert-level verification accuracy at less than 20% of the time cost.

## I   FUTURE DIRECTIONS

While SpeechQC-Agent demonstrates strong performance in synthetic and vendor datasets, several avenues remain open to advancing both scope and methodology. First, we aim to develop **SpeechQC-Agent+**, a dynamic extension capable of synthesizing new QC tasks on the fly from quires from natural language, enabling adaptation to emerging dataset standards and modalities. Second, to address current gaps such as dialect imbalance detection and speaker bias auditing, we plan to integrate clustering-based dialect tagging, metadata-speech alignment, and distributional fairness checks. Third, scaling to 100k hour corpora requires distributed execution and schedul-

ing optimizations; we will explore hybrid orchestration strategies combining LLM planners with lightweight rule-based controllers to reduce cost and latency. Fourth, we envision extending the framework beyond speech to multimodal data curation, including OCR datasets, vision-language corpora, and conversational agents, positioning dataset verification as a unified agentic AI problem. Finally, incorporating formal guarantees on workflow soundness and hallucination containment, e.g., using schema validation, consistency checks, and LLM-as-judge ensembles, offers a path toward provably reliable agentic QC systems.

## J  LIMITATIONS

Although SpeechQC-Agent represents a first step toward scalable quality verification of multilingual speech datasets, it has several limitations. First, the performance of the system strongly depends on **PlannerLLM**. Smaller models (e.g., LLaMA-3.1-8B) underperform due to limited exposure to Indic languages, and even stronger models (e.g., GPT-4.1-mini) sometimes hallucinate additional steps, creating reliability challenges. Second, the current evaluation focuses on the verification of **audio-transcript**; extensions to multimodal data (e.g. speech-translation or dialogue corpora) remain unexplored. Third, the system cannot yet detect **dialect/accent imbalance** or **socio-linguistic biases** in large corpora. Finally, runtime remains non-trivial for large corpora, and while significantly faster than human annotation, scaling to hundreds of thousands of hours will require additional optimization and distributed execution strategies.

Despite these limitations, SpeechQC-Agent establishes a foundation for future research in dynamic, LLM-driven quality control of speech and multimodal datasets.

## K  PROMPTS

## Task Selection Prompt

**Prompt:**
You are given the following functions:
1. ASR Transcription
2. Number of Speakers calculation and duration per speaker
3. Quality of Transcript
4. Graphene or character calculation
5. Vocab calculation
6. Language identification
7. Audio length calculation
8. Silence calculation (using VAD)
9. Sample rate check
10. CTC score calculation
11. Upsampling Check
12. Check if speakers are new or old
13. Check the domain of the speech dataset
14. Map transcriptions to audio files using forced alignment
15. Language identification using ASR transcriptions and IndicLID
16. Normalization by removing HTML and other tags from transcriptions in JSON or XML files
17. Evaluate transcript coherence and fluency using LLM-as-a-Judge and score out of 10
18. Transliteration - Convert Roman script words to Native script using Transliteration for a specified file and language

Based on the prompt, reply with task numbers that have to be done without any explanation or reasoning.

**Input:**
Prompt: {user_prompt}

**Output Format:**
Example: 1,3,5

## Topological Sorting Prompt

**Prompt:**
You are given the following functions:
1. ASR Transcription using audio files
2. Number of Speakers calculation and duration per speaker using audio files
3. Quality of Transcript using transcriptions
4. Graphene or character calculation using transcriptions
5. Vocab calculation using transcriptions
6. Language identification using transcriptions
7. Audio length calculation using audio files
8. Silence calculation (using VAD) using audio files
9. Sample rate check using audio files
10. CTC score calculation using audio files and transcriptions
11. Upsampling Check using audio files
12. Check if speakers are new or old using the results from number of speakers calculation
13. Check the domain of the speech dataset using transcriptions from ASR
14. Map transcriptions to audio files using forced alignment, using ground truth transcriptions
15. Language identification using ASR transcriptions and IndicLID, using transcriptions from ASR
16. Normalization by removing HTML and other tags from transcriptions in JSON or XML files
17. Evaluate transcript coherence and fluency using LLM-as-a-Judge and score out of 10, using transcriptions from ASR
18. Transliteration - Convert Roman script words to Native script using Transliteration, using a specified file and language code from the prompt

We have to do tasks: {resp_1}.

Make a Topological sorting for what is the best way to proceed with these tasks, sequentially and concurrently.

**Guidelines:**
- We can do tasks concurrently if they are independent of each other.
- Task 12 depends on task 2.
- Task 13 depends on task 1.
- Task 14 depends on the ground truth conversion process.
- Task 15 depends on task 1.
- Task 17 depends on task 1.
- Task 18 is independent.

**Output Format:**
Example: [[1,3], [5], [8]] (this means do 1 and 3 concurrently, then do 5, and finally do 8)

Finally, give me the topological sorting for the tasks: {resp_1} without any explanation or reasoning.

**Input Source Determination Prompt**

**Prompt:**
Determine the source of the following inputs for task {task_id}:
{json.dumps(required_inputs, indent=2)}

**Parameters:**
Possible sources:
- User prompt: {state.get('user_prompt', '')}
- Previous task outputs in CombinedStateDict: {json.dumps(k: v for k, v in state.items() if k not in ['folder_path', 'user_prompt', 'execution_log', 'task_inputs', 'topological_sort'], indent=2)}
- Default: folder_path={state.get('folder_path', '')}

**Output Format:**
Return a JSON object mapping each input to its source value or an error message if not found.

**Corruption Check Prompt**

**Prompt:**
You are given a folder with audios at this path: {state['folder_path']}.

Write a Python script to:
- Attempt to open and read each audio file.
- If a file fails to load or raises an error, mark it as corrupted and capture the error message.

Save a CSV listing all files and their status ("Corrupt" or "Valid") as audio_validity.csv in the same directory.

Finally, Respond with "Success" if all files are valid, otherwise "Invalid".

**Audio Extension and Format Check Prompt**

**Prompt:**
You are given a folder with audios at this path: {state['folder_path']}.

Write a Python script to:
1. Confirm that each file except {file_path} has a valid audio extension (only .wav or .mp3). Ignore files with extensions: .csv, .xml, and .json (do not process, validate or flag them).
2. For audio files, also check if they are in WAV format by attempting to read them using a library like wave or librosa.
3. Create a CSV with columns: Filename, Valid_Extension, Is_WAV_Format, Status
4. Status should be "Pass" only if both extension is valid and format is WAV.
5. Save the CSV as audio_format_check.csv in the same directory.

Respond with "Success" if all files pass, otherwise "Invalid".

## Sample Rate Check Prompt

**Prompt:**
You are given a folder with audio files at this path: {state['folder_path']}.

Write a Python script to:
1. Check each audio file's sample rate
2. Create a CSV with columns: Filename, Sample_Rate, Status
3. Store "Pass" in Status if sample rate is 16000 Hz, otherwise "Fail"
4. Save the CSV as `sample_rate_check`.csv in the same directory

Use libraries like librosa, soundfile, or wave to check the sample rate.

## Ground Truth File Conversion Prompt

**Prompt:**
You are given a file of ground truths of audios {state['folder_path']} at {file_path}.

1. Get the structure of the txt, csv, json, xml file.
2. Identify the element/column that contains the filename and transcriptions (ground truth). If there is no such column, return "Invalid".
3. Convert the file to CSV with added columns of Filename and Transcription.
4. Save the updated CSV with the new column to the same directory as new_transcriptions.csv.

Finally, Respond with "Success" if all steps are done, otherwise "Invalid".

## Conversation Generation Prompt

**Prompt:**
You are a conversation generator tasked with creating realistic dialogue between exactly two speakers in English.
Topic: {topic}
Setting: {setting}
Speakers: {speaker1} and {speaker2}

**Requirements:**
- The conversation must be rich in content related to the specified topic and reflect the given setting.
- Generate a long conversation with approximately 100 dialogue exchanges.
- Format the output strictly as:
{speaker1}: sentence1
{speaker2}: sentence2
{speaker1}: sentence3
...and so on.
- Do not include any explanations, actions, or additional text outside the conversation format.
- Ensure the conversation flows naturally and is meaningful with detailed exchanges relevant to the setting and topic.

 **Output:**

**Translation Prompt**

**Prompt:**
Translate the following sentence into {language} while maintaining realism and natural flow.
**Guidelines:**
- The conversation should primarily be in {language}, but preserve certain English words commonly used by {language} speakers.
- Enclose all preserved English words within ¡eng¿...¡/eng¿ tags.
- Randomly and sparsely insert conversational effect tags such as [babble], [bg-speech], [laugh], [music], [no-speech], [noise], [overlap], or [silence].
- Use ¡initial¿...¡/initial¿ tags for any initials or abbreviations.
- Avoid overusing English words and tags; include them only when contextually appropriate.
- Output only the translated sentence without any explanation.

**Input:**
Sentence: {content}

**Output Format:**
Translation: [Translated sentence will be provided here in the specified format with appropriate tags.]

**Conversation Metadata Prompt**

**Prompt:**
Generate conversation metadata based on the provided conversation content.
**Input:**
Conversation: {translated_content}

**Output Format:**
Generate conversation metadata in the following JSON format:
{"domain":"¡domain¿","topic":"¡topic¿","language":"{language}","conversation_name":"{conv_id}-GPT"}

**Instructions:**
- Determine the "domain" and "topic" based on the conversation content.
- Set "language" to the predominant language of the conversation.
- Use the provided "conversation_name" as is.
- Provide only the raw JSON string without any explanation or formatting wrappers.

1512
1513
1514
1515
1516
1517
1518
1519
1520
1521
1522
1523
1524
1525
1526
1527
1528
1529
1530
1531
1532
1533
1534
1535
1536
1537
1538
1539
1540
1541
1542
1543
1544
1545
1546
1547
1548
1549
1550
1551
1552
1553
1554
1555
1556
1557
1558
1559
1560
1561
1562
1563
1564
1565

### Speaker Details Prompt

**Prompt:**
Generate speaker information for two speakers based on the provided conversation content.
**Input:**
Conversation: {translated_content}

**Output Format:**
Generate speaker information for {speaker1} and {speaker2} in the following JSON format:
```
{
"{speaker1}": {
"speakers": [
{
"gender": "¡male or female¿",
"speakerId": "¡alphanumeric ID¿",
"recorderId": "¡alphanumeric ID¿",
"nativity": "{language}",
"ageRange": "¡age range like 25-34¿"
}
]
},
"{speaker2}": {
"speakers": [
{
"gender": "¡male or female¿",
"speakerId": "¡alphanumeric ID¿",
"recorderId": "¡alphanumeric ID¿",
"nativity": "{language}",
"ageRange": "¡age range like 35-44¿"
}
]
}
}
```

**Instructions:**
- Follow the exact JSON structure shown above with all opening and closing braces properly matched.
- Randomly assign values for "gender" (choose either "male" or "female").
- For "speakerId", use a format like "S-XXXXX" where X is a digit.
- For "recorderId", use a format like "RXXX" where X is a digit.
- Set "nativity" to exactly "{language}" as provided.
- For "ageRange", use one of these formats: "18-24", "25-34", "35-44", "45-54", "55-64", "65+".
- Ensure the JSON is properly formatted and valid - all quotes, commas, and braces must be correctly placed.
- Provide only the raw JSON string without any explanation, markdown formatting, or code blocks.

1566
1567
1568
1569
1570
1571
1572
1573
1574
1575
1576
1577
1578
1579
1580
1581
1582
1583
1584
1585
1586
1587
1588
1589
1590
1591
1592
1593
1594
1595
1596
1597
1598
1599
1600
1601
1602
1603
1604
1605
1606
1607
1608
1609
1610
1611
1612
1613
1614
1615
1616
1617
1618
1619

**Transcription Function Prompt**

**Prompt:**
Transcribe audio files from a specified folder and return the transcription output in CSV format. This task assumes that all audio files are in Hindi.
**Input:**
- A folder path containing audio files.
- The folder must exist and be a valid directory.
- All audio files should be in Hindi.

**Output Format:**
A dictionary with the following structure:
{
"A" [where A is node in the node graph]: "¡CSV transcription result or error message¿",
"audio_dir": "¡Path to the input folder¿"
}

**Instructions:**
- Validate that the provided folder path exists and is a directory.
- If invalid, return the error message: `"A": "Error: Invalid audio directory"`.
- If valid, perform transcription of all audio files in the folder.
- Use the `transcribe_folder_to_csv()` function for transcription.
- Assume the source language is "Hindi".
- Log the transcription process using appropriate logging levels (info and error).
- Return the transcription results in the key "A" along with the input directory.

**Silence Detection Prompt**

**Prompt:**
Perform silence detection on all audio files within a specified directory and return the result.
**Input:**
- A directory path containing audio files to be processed.
- The folder must exist and be a valid directory.

**Output Format:**
A dictionary with the following structure:
{
"D": "¡Silence detection result or error message¿"
}

**Instructions:**
- Check if the provided audio directory exists and is valid.
- If the directory is invalid or not found, return the error message: `"D": "Error: Invalid audio directory"`.
- If valid, apply silence detection to all audio files in the directory using the `process_folder_vad()` function.
- Log the beginning of the detection process with an info-level message.
- Return the result under the key "D".

**Vocabulary Extraction Prompt**

**Prompt:**
Extract unique words (vocabulary) from the transcriptions in a CSV file and save them into a new column. Output the updated CSV with the extracted vocabulary.
**Input:**
- A directory containing a CSV file, typically named `indicconf_hypothesis.csv`.
- The CSV must have a column named `Transcription` or `Ground_Truth` (case-insensitive).

**Output Format:**
A dictionary in the following format:
{
"vocab_output": "¡Path to vocab_list.csv or error message¿"
}

**Instructions:**
- Locate the CSV file using the key `"A"` in state, or fallback to `audio_dir/indicconf_hypothesis.csv`.
- If the file doesn't exist, return: `"vocab_output": "Error: CSV file <path> not found"`.
- Within the CSV, identify the transcription column by searching for 'Transcription' or 'Ground_Truth' (case-insensitive).
- For each row, extract a list of **unique words** from the transcription.
- Store the list in a new column named `vocab_list`.
- Save the updated CSV as `vocab_list.csv` in the same directory.
- Return `"vocab_output": "CSV saved at: <path>"` if successful.
- If the agent fails to complete the task or the file is not created, return an appropriate error message.
- Handle and log all exceptions clearly.

**Character Extraction Prompt**

**Prompt:**
Extract unique characters from each transcription in a CSV file and save them into a new column. Output the updated CSV with the extracted characters.
**Input:**
- A directory containing a CSV file, typically named `indicconf_hypothesis.csv`.
- The CSV must have a column named `Transcription` or `Ground_Truth` (case-insensitive).

**Output Format:**
A dictionary in the following format:
{
"character_output": "¡Path to character_list.csv or error message¿"
}

**Instructions:**
- Locate the CSV file using the key `"A"` in state, or fallback to `audio_dir/indicconf_hypothesis.csv`.
- If the file doesn't exist, return: `"character_output": "Error: CSV file <path> not found"`.
- Identify the transcription column by searching for 'Transcription' or 'Ground_Truth' (case-insensitive).
- For each row, extract a list of **unique characters** from the transcription.
- Store the list in a new column named `character_list`.
- Save the updated CSV as `character_list.csv` in the same directory.
- If the script completes successfully and the file is created, return: `"character_output": "CSV saved at: <path>"`.
- If the agent fails or the output file is not found, return an appropriate error message.
- Log any exceptions during processing clearly and accurately.

1728
1729
1730
1731
1732
1733
1734
1735
1736
1737
1738
1739
1740
1741
1742
1743
1744
1745
1746
1747
1748
1749
1750
1751
1752
1753
1754
1755
1756
1757
1758
1759
1760
1761
1762
1763
1764
1765
1766
1767
1768
1769
1770
1771
1772
1773
1774
1775
1776
1777
1778
1779
1780
1781

---

**Audio Length Calculation Prompt**

**Prompt:**
Calculate the duration of each audio file in a given folder and save the results in a CSV file.
**Input:**
- A valid directory path containing audio files.

**Output Format:**
A dictionary in the format:
{
"audio_length_output": "¡Result of operation or error message¿"
}

**Instructions:**
- Check if the `audio_dir` exists and is a directory. If invalid, return: `"audio_length_output"`: `"Error: Invalid audio directory"`.
- Write a Python script that performs the following tasks:
1. Iterate over all audio files in the directory.
2. Calculate the duration of each audio file in seconds.
3. Store the filename and corresponding duration in a CSV with columns: `Filename`, `Audio_length`.
4. Save the resulting CSV as `audio_length.csv` in the same folder.
- Execute the script using the `[python_repl]` tool.
- Return the script's output message under the key `"audio_length_output"`.
- In case of failure or exceptions, return an appropriate error message.
- Log errors clearly to aid debugging.

1782
1783
1784
1785
1786
1787
1788
1789
1790
1791
1792
1793
1794
1795
1796
1797
1798
1799
1800
1801
1802
1803
1804
1805
1806
1807
1808
1809
1810
1811
1812
1813
1814
1815
1816
1817
1818
1819
1820
1821
1822
1823
1824
1825
1826
1827
1828
1829
1830
1831
1832
1833
1834
1835

### Devanagari Script Verification Prompt

**Prompt:**
Verify whether each transcription in a CSV file is written in the Devanagari script using Unicode checks.
**Input:**
- Path to a CSV file (e.g., `indicconf_hypothesis.csv`) with a column containing ground truth text.

**Output Format:**
A dictionary in the format:
{
"language_verification_output": "¡Result of operation or error message¿"
}

**Instructions:**
- Load the CSV file and identify the transcription column (case-insensitive: 'Ground_Truth', 'Transcription', etc.).
- For each row:
1. Remove whitespace and punctuation from the transcription.
2. Check if all remaining characters fall within the Unicode range U+0900-U+097F (Devanagari script).
3. If they do, set `Is_Devanagari` to `True`; otherwise `False`.
4. If the transcription is empty or only punctuation, set `Is_Devanagari` to `False`.
- Add a new column `Is_Devanagari` to the CSV.
- Save the output file as `language_verification.csv` in the same directory.
- Ensure the final CSV includes: `Filename`, `Transcription`, `Is_Devanagari`.
- Use the `[python_repl]` tool to execute the script.
- On success, return `"Success"`; else provide an error message.
- Handle edge cases and log any errors encountered.

**CTC Score Computation Prompt**

**Prompt:**
Compute Connectionist Temporal Classification (CTC) alignment scores from audio-transcription pairs and classify alignment quality.
**Input:**
- A directory containing audio files (`audio_dir`)
- A CSV file (e.g., `indicconf_hypothesis.csv`) with aligned transcripts, identified via key `'A'`

**Output Format:**
A dictionary in the format:
{
"ctc_score_output": "¡CSV output path or error message¿"
}

**Instructions:**

- Load the CSV and audio directory.
- For each audio file, compute alignment scores using the transcriptions in the CSV.
- Use `process_audio_directory()` to return segment-wise alignment with scores and timestamps.
- Aggregate results by:
  - Grouping by filename.
  - Combining the segment labels into a full transcript (`Aligned_Transcript`).
  - Taking the average CTC score as `CTC_Score`.
  - Serializing segment-level details (label, start, end, score) into JSON under `Aligned_Segments`.
- Classify the score using:
  - `Good` if score > 0.7
  - `Medium` if score > 0.5
  - `Poor` otherwise
- Save the final CSV with columns: `Filename`, `Aligned_Segments`, `Aligned_Transcript`, `CTC_Score`, `CTC_Status`.
- Output the result to `ctc_scores.csv` in the same directory as the input CSV.
- Log and report errors appropriately.

**Valid Speaker Verification Prompt**

**Prompt:**
Analyze speaker presence across files to determine whether a speaker is "New" or "Old" based on repetition across files.

**Input:**
- A directory containing a CSV named `num_speakers.csv` with columns:

- `File Name`
- `Number of Speakers`
- `Speaker Durations` - JSON object mapping speaker IDs to durations

**Output Format:**
A dictionary:
{
"valid_speaker_output": "¡CSV output path or error message¿"
}

**Instructions:**

1. Load `num_speakers.csv`.

2. Build a dictionary to track how many files each speaker appears in.

3. For each row:

- Skip if `Number of Speakers` == "Error".
- If only one speaker and `SPEAKER_00` is reused across files, mark as `Old`.
- If multiple speakers and any speaker is reused across files, mark as `Old`.
- Otherwise, mark the speaker as `New`.

4. For each row, populate:

- `Filename`
- `Speaker_Status` (`New` or `Old`)
- `Common_File` (the current file name if status is `Old`, else empty)

5. Save the result to `valid_speaker.csv` in the same directory.

6. Respond with "Success" if the script runs without errors and file is saved. Otherwise, return "Invalid".

### Domain Checker Prompt

**Prompt:**
You are a Hindi language expert. Analyze the following normalized Hindi transcript and determine the general domain of the speech dataset.
**Instructions:**

- Return the domain as a **single word** (e.g., `News`, `Call Center`, `Interview`, `Conversation`, `Education`).

**Input:**
A CSV file `indicconf_hypothesis.csv` located inside a directory, containing a column named `transcriptions` with normalized Hindi transcripts.
**Expected Output:**
A new column `domain` added to the CSV, representing the predicted domain of each transcription. The final output is saved as `domain_check.csv` in the same directory.
**Agent Behavior:**

1. Validate the input directory and CSV.

2. Iterate over each row in the `transcriptions` column.

3. For each transcript, send a prompt to the language model to classify the domain.

4. If the LLM fails, label the domain as `Unknown`.

5. Save the resulting DataFrame with the new `domain` column to `domain_check.csv`.

### IndicLID Language Identification Agent Prompt

**Prompt Objective:**
Identify the language of each transcript using the IndicLID model.
**Input Description:**

- A folder containing a CSV file (default name: `indicconf_hypothesis.csv`).

- The CSV should include a column named `transcriptions` and optionally `Filename`.

**Instructions:**

1. For each row in the CSV:

    - Extract the transcript and filename.
    - If the transcript is empty or NaN, assign `Language_Code = Unknown`, `Confidence = 0.0`, `Model_Used = IndicLID`.
    - Otherwise, use the IndicLID model to perform language identification.

2. If language identification fails for a transcript, mark it with `Language_Code = Error`.

3. Store all results in a new DataFrame with columns: `Filename`, `Transcription`, `Language_Code`, `Confidence`, `Model_Used`.

4. Save the output as `indiclid_language_identification.csv` in the same directory.

**Expected Output:**
A CSV file containing language identification results for each transcript, with confidence scores and the model used (`IndicLID`).

## Text Normalization and Tag Removal Agent Prompt

**Prompt Objective:**
Normalize transcription text by cleaning ground truth data in a CSV file.
**Input Description:**

- A directory containing a CSV file named `indicconf_hypothesis-gt.csv`.
- The file should have a column named `Transcriptions` or `ground_truth` (case-insensitive).

**Instructions:**

1. Read the CSV file and identify the transcription column (`Transcriptions` or `ground_truth`).
2. Clean each transcript using the following rules:
   - Remove HTML tags like `` and ``.
   - Remove any text enclosed in square brackets (e.g., `[START]`).
   - Remove symbols such as `#`, `$`, and `%`.
3. Add a new column named `normalized_transcripts` with the cleaned text.
4. Save the updated CSV as `normalized_list.csv` in the same directory.

**Expected Output:**
A new CSV file with the original columns and an additional `normalized_transcripts` column saved as `normalized_list.csv`.

## LLM-Based Transcription Quality Scoring Agent Prompt

**Prompt Objective:**
Evaluate the fluency and coherence of ASR-generated transcriptions using a Language Model (LLM) and assign scores and comments.
**Input Description:**

- A directory path containing a CSV file named `indicconf_hypothesis.csv`.
- The CSV contains:
  - `Filename` column (case-insensitive).
  - One of `ground_truth` or `transcriptions` columns (case-insensitive), containing ASR outputs.

**Instructions:**

1. Load the CSV file.
2. For each transcription:
   - Analyze sentence fluency and meaning very strictly.
   - Score each transcription from **0 to 10**:
     - 10: Highly meaningful and fluent Hindi sentence.
     - 0: Nonsensical or contains language other than Hindi.
     - Gradually decrease score based on fluency degradation.
   - Provide a brief `Evaluation_Comment` justifying the score.
3. Create a new CSV file with the columns: `Filename`, `Transcription`, `LLM_Score`, and `Evaluation_Comment`.
4. Save the output as `llm_scores.csv` in the same directory.
5. Handle errors gracefully during execution.

**Expected Output:**
A CSV file named `llm_scores.csv` containing scored and reviewed transcriptions.

2052
2053
2054
2055
2056
2057
2058
2059
2060
2061
2062
2063
2064
2065
2066
2067
2068
2069
2070
2071
2072
2073
2074
2075
2076
2077
2078
2079
2080
2081
2082
2083
2084
2085
2086
2087
2088
2089
2090
2091
2092
2093
2094
2095
2096
2097
2098
2099
2100
2101
2102
2103
2104
2105

---

**English Word Count Agent Prompt**

**Prompt Objective:**
Determine the number of English words present in each line of a normalized transcript using an LLM.

**Input Description:**

- A directory path that contains a CSV file named `normalized_list.csv`.
- The CSV must have a column named `ground_truth`, containing the transcription text.

**Instructions:**

1. Load the `normalized_list.csv` file.
2. For each row in the `ground_truth` column:
   - Construct a prompt asking a language expert to count the number of English words (case-insensitive) in the given text.
   - Extract the integer response.
   - If the LLM fails, assign $-1$ for that row.
3. Append the count as a new column called `english_word_count`.
4. Save the updated CSV as `english_word_count.csv` in the same directory.

**Prompt Template:**

You are a language expert. Count and return only the number of **English words** (case-insensitive) in the following text.
Text:
{ground_truth_text}
Respond with just the number.

**Expected Output:**
A CSV named `english_word_count.csv` containing an additional column `english_word_count` with English word frequencies per row.

**Utterance Duplicate Checker Agent Prompt**

**Prompt Objective:**
Identify and report duplicate utterances across all text-based columns in a CSV.
**Input Description:**

- A directory containing a CSV file named `normalized_list.csv`.

**Instructions:**

1. Load the `normalized_list.csv` file.
2. Iterate through each column of the DataFrame.
3. For columns with text (`dtype == object`):
   - Detect duplicated utterances (preserve all duplicates using `keep=False`).
   - For each unique duplicated utterance, count the number of occurrences.
   - Record the column name, the duplicated utterance, and the count.
4. Save the results in a new CSV called `duplicate_utterances.csv` containing:
   - `column_name`, `utterance`, and `count`
5. If no duplicates are found, return a message indicating that.

**Expected Output:**

- A CSV file named `duplicate_utterances.csv` if duplicates exist.
- Otherwise, a message stating `"No duplicate utterances found."`

---

**WER Computation Agent Prompt**

**Prompt Objective:**
Compute Word Error Rate (WER) between normalized reference transcriptions and predicted hypotheses.
**Input Description:**

- A directory containing two CSV files:
  - `normalized_list.csv` with the column `normalized_transcripts`.
  - `indicconf_hypothesis.csv` with the column `transcriptions`.

**Instructions:**

1. Ensure both CSVs exist and contain the same number of rows.
2. For each row, compute the Word Error Rate (WER) between:
   - Reference ← `normalized_transcripts`
   - Hypothesis ← `transcriptions`
3. Use the `jiwer` library for WER calculation.
4. Handle exceptions on a per-row basis to ensure continuity even if some rows fail.
5. Save the output in a CSV named `wer.csv` with columns:
   - `Reference`, `Hypothesis`, and `WER`

**Expected Output:**

- A CSV file named `wer.csv` saved in the same directory.
- Each row shows the WER score for the respective transcription pair.

2160
2161
2162
2163
2164
2165
2166
2167
2168
2169
2170
2171
2172
2173
2174
2175
2176
2177
2178
2179
2180
2181
2182
2183
2184
2185
2186
2187
2188
2189
2190
2191
2192
2193
2194
2195
2196
2197
2198
2199
2200
2201
2202
2203
2204
2205
2206
2207
2208
2209
2210
2211
2212
2213

---

**Graph Builder Agent Prompt**

**Prompt Objective:**
Construct a 'StateGraph' from a structured list of task groups while filtering by a valid task set.

**Input Description:**

- `structure`: A list of lists where each sublist represents a group of task IDs that can be executed in parallel.

- `valid_tasks`: A set of valid task identifiers (as strings). Only these will be included in the final graph.

**Instructions:**

1. Filter the `structure` to retain only task IDs present in `valid_tasks`.

2. If the resulting structure is empty but `valid_tasks` is non-empty, use all numeric valid tasks as a fallback.

3. Add each valid task as a node in the graph using `node_map`, which maps `task_id` to a tuple: (`node_name, function, description`).

4. Add a dummy `start` node and connect it to the first group.

5. Connect each group to the next group, allowing fan-in/fan-out connections.

6. Connect the last group to the terminal `END` node.

**Expected Output:**

- A compiled `StateGraph` object that respects the dependency structure implied by the groupings and task validity.

- An error is raised if no valid tasks remain after filtering.

## Prompt Checker Agent Prompt

**Prompt Objective:**
Analyze a user's natural language prompt to determine whether the currently selected task IDs are appropriate, and update the task list if any are missing based on defined task descriptions and selection rules.

**Input Description:**

- `user_prompt`: A natural language prompt provided by the user describing the task they want to perform.
- `selected_tasks`: A comma-separated string of task numbers (e.g., "1,2,5") that have been initially selected for execution.

**Task Descriptions:**

- Contains 24 predefined task definitions, ranging from ASR transcription to WER computation.

**Selection Rules:**

- Uses keyword and semantic rules (e.g., "if prompt mentions 'Vocab calculation', include task 5") to guide inclusion.
- Tasks 1 and 15 are linked if language identification is mentioned.
- Certain tasks (e.g., 9, 23, 24) trigger the inclusion of dependent tasks (e.g., task 16).

**Instructions to the LLM:**

1. Analyze the `user_prompt` and determine which tasks are required based on semantic understanding and rules.
2. Compare the determined tasks with `selected_tasks`.
3. If any tasks are missing, return `Status: Missing`, with task IDs and an explanation.
4. If all are correct, return `Status: Correct` and the list of tasks.
5. Format the output as:

```
Status: <Correct|Missing>
Tasks: <comma-separated task IDs>
Explanation: <why tasks were added (if Missing)>
```

**Execution Loop:**

- Repeats for a maximum of 3 iterations to ensure task completeness.
- Dynamically updates task list with each LLM feedback.
- Calls `select_tasks()` if new insights are needed.

**Output:**

- Returns the final list of task IDs as a comma-separated string.

