# OpenReview forum: "SpeechQC-Agent: A Natural Language Driven Multi-Agent System for Speech Dataset Quality"
_ICLR.cc/2026/Conference — ICLR 2026 Conference Withdrawn Submission_

### Official Review · Reviewer_bkFa · 2025-10-29

**Soundness:** 1
**Presentation:** 2
**Contribution:** 1
**Rating:** 2
**Confidence:** 4

**Summary:**

SpeechQC-Agent proposes a multi-agent LLM system that converts natural language quality control requirements into executable workflows for speech dataset validation. The system uses DAG-based task decomposition, combines LLM-synthesized and pre-defined tools, and evaluates 5 LLM planners across 24 verification tasks. While addressing a genuine practical problem, the paper suffers from significant methodological weaknesses and insufficient experimental rigor that prevent acceptance at ICLR in its current form.

**Strengths:**

1. Addresses Real Problem with Clear Motivation: Speech dataset quality control is genuinely expensive and time-consuming.
2. Novel Application Domain for Multi-Agent Systems: This appears to be the first integrated system combining natural language-driven workflow generation with multi-agent orchestration specifically for speech dataset quality control.
3. New Benchmark Contribution: SpeechQC-Dataset

**Weaknesses:**

1. Synthetic Benchmark Validity (Critical): Controlled perturbations ≠ real-world complexity. Prior research shows synthetic benchmarks often underestimate task difficulty and yield rankings inconsistent with real-world evaluations. The authors should validate their data by training speech processing systems on it and comparing performance with systems trained on real-world data.

2. Missing Baseline: No human expert baseline is provided. Where are multiple expert annotations with inter-rater agreement (e.g., Cohen’s kappa, Krippendorff’s alpha)? How is “expert-level” defined? There’s also no comparison with existing tools or ablations against single-agent LLM or simple prompt-engineering baselines.

3. Limited Applicability: Evaluation is restricted to a single language (Hindi) and only 15 hours of generated data—less than the amount available in existing benchmarks (e.g., OpenSLR 118 https://www.openslr.org/118/).

4. Limited Novelty: The contribution is primarily system integration of existing components rather than any substantive algorithmic innovation.

**Questions:**

Check weaknesses

---

### Official Review · Reviewer_aioD · 2025-10-29

**Soundness:** 2
**Presentation:** 3
**Contribution:** 2
**Rating:** 4
**Confidence:** 3

**Summary:**

The paper introduces SpeechQC-Agent, a natural language-driven multi-agent system for large-scale speech dataset quality control. It addresses the limitations of static, domain-specific QC pipelines by leveraging a LLM planner to decompose user queries into dynamic directed acyclic graph workflows. The system utilizes modular sub-agents that execute specific QC tasks and features an iterative, LLM-guided task selection and refinement loop to ensure completeness. From a multimodal perspective, the framework implicitly performs a high-level fusion of control and domain-specific tools, allowing human intent to govern the orchestrated execution of modality-specific analysis tasks.

**Strengths:**

S1) The concept of a natural language-driven, generalized, multi-agent framework for data QC can be employed for reducing the dependence of human experts.

S2) Employing modular sub-agents and a DAG execution structure facilitates easy incorporation of new QC tools and enables generalization across different modalities, vendors, and languages.

S3) The system demonstrates a successful fusion of the LLM's symbolic reasoning/planning modality with the computational execution modality.

**Weaknesses:**

W1) This study is vague on how the outputs of the various sub-agents are ultimately fused and synthesized into a final QC report. Are they merely appended sequentially, or is there a subsequent LLM step for semantic fusion/summary?

W2) Depending on the LLM for up to three iterations of task refinement introduces significant latency and computational cost. This can hinder adoption for real-time or massive-scale QC applications.

W3) The structure seems focused on workflow. Explicit discussion of how fine-grained modality analysis is integrated beyond simple ASR or WER tasks is missing.

**Questions:**

Q1) How is the final QC result generated from the multiple sub-agent outputs? Is there a dedicated Fusion Agent responsible for synthesizing disparate metrics into a single, cohesive human-readable quality score or report?

Q2) Could the authors elaborate on which of the 24 predefined tasks specifically fall under fine-grained modality analysis versus simple data-level computation?

Q3) In a scenario where two different sub-agents produce conflicting results, how does the LLM planner or the execution loop manage this conflict before finalizing the DAG output?

---

### Official Review · Reviewer_ouXL · 2025-11-01

**Soundness:** 2
**Presentation:** 2
**Contribution:** 2
**Rating:** 2
**Confidence:** 4

**Summary:**

This paper presents SpeechQA-Agent, which is an agent system that intends to conduct speech data quality verification automatically.

The agent would first understand the user's intention, making it into an action list. The action list is then transformed into a graph and conducted in parallel.

In experiments, the authors report the success rate of the design with varying LLMs. The authors claim the proposed agent can improve work efficiency.

**Strengths:**

The reviewer appreciates the authors' intention to design an agent to conduct speech data verification. It also proposes a theoretically feasible workflow for such an intention.

**Weaknesses:**

I would appreciate it if the reviewers could respond to the following concerns:

(1) It is claimed "SpeechQC-Agent achieves 80-90% of expert-level accuracy while operating at less than 20% of the cost and time" in the abstract, but I cannot find the support material in the main content. Section 5.3 also mentioned "Human Annotation baseline", which cannot be found in Tables 2,3,4,5. The paper also doesn't compare with any prior agent baseline methods, or the previous "static, domain-specific, and heavily reliant on human experts" method.

(2) Although Table 1 mentions many different check items, Tables 2 and 3 only report these items selectively.

(3) There is considerable duplication between Section 3 and Section 5.1

(4) For the proposed SpeechQC-dataset, many details are missing. E.g., although the data could contain various flaws (which the agent intends to detect), there are no statistics about the distribution of each error category. The whole data simulation pipeline leverages many tools that could contain error propagation, such as a multilingual LLM for low-resource languages and TTS. These errors should be properly handled and analyzed when building a benchmark (as this benchmark is used to justify the effectiveness of the proposed agent)

(5) Tables 4 and 5 compare various LLM planning capabilities. Although this is important to an agent, such capability is intrinsic to the LLMs and is not a contribution to this work. Instead, the author claims that the parallel planning (together with the graph workflow) is a major benefit of this agent, but didn't provide experimental support for it.

**Questions:**

As in weakness.

---

### Official Review · Reviewer_FF7f · 2025-11-02

**Soundness:** 2
**Presentation:** 1
**Contribution:** 2
**Rating:** 2
**Confidence:** 3

**Summary:**

The paper proposes SpeechQC-agent, a natural language driven agent for speech dataset quality control. Given a natural language user input, the agent produces code that checks the quality of paired speech data, such as audio length, sample rate, normalization, and domain classification. The authors test the agent using SpeechQC-Dataset, a synthetic benchmark developed using LLM-driven dialogue generation, translation, and synthesized into speech via TTS. Results suggest that SpeechQC-agent can achieve strong verification accuracy while being more efficient than human annotation.

**Strengths:**

- This paper presents a novel method for speech dataset qc using LLM agents
- The authors evaluated different ways to implement agent-based speech qc, showing the weakness of single-agent systems

**Weaknesses:**

- Parts of the paper are difficult to understand. Figure 3 is too dense to be easily parse and its difficult to capture what the numbers in the results tables are measuring / intended to convey. Minor formatting issues such as the inconsistent use of sig figs in each row.
- I am not convinced about the usefulness of the proposed method. While a general speech QC agent would indeed be useful for open-ended tasks (like coherence evaluation) , it seems excessive to employ one to simply check the sample rate of an audio file. Considering how dense some of these prompts are for extremely simple programs (like `Sample Rate Check Prompt` and `Audio Length Calculation Prompt`), why rely on an agentic framework that must re-write the code every time and not just call a pre-written function? Most of the evaluated tasks are not open ended and can be done with static pre-written functions. The results in Table 2 even indicate that the agents often fail to even perform these operations. Using the agent to call pre-written functions should perform just as well while being more efficient.

**Questions:**

- For a paper that focuses primarily on low-resource languages, it is quite surprising that there is no mention of this aspect in the title and abstract, making the intro seem very disjoint from the above elements.
- Similarly, why not just stress-test verification systems directly on English? All the synthetic dialogues are originally generated in English anyways. Using LLMs to translate to low-resource languages (and then synthesizing into speech with TTS) increases the risk of hallucination, which risks making the shown results less reliable.
- Missing citation in line 82 after "planning"?
- In Table 4, what are QC1-1, 1-2, and 1-3? Why only evaluate on 9 audio samples total if the benchmark set has thousands of examples?
- I do not understand what the authors are trying to show in Table 3 and the caption is unhelpful. Is it measuring the accuracy of the model in detecting Roman characters, WER of the synthetic audio (does 0.12 mean 12% WER here or 0.12%?), and if the counted number of HTML tags is correct?

---

### Note · Authors · 2025-12-21

**Comment:**

I want to thank all the reviewer for their effort.

**Withdrawal Confirmation:**

I have read and agree with the venue's withdrawal policy on behalf of myself and my co-authors.